# HERA: a high-resolution pan-European hydrological reanalysis (1951-2020)

Aloïs Tilloy[1], Dominik Paprotny[2], Stefania Grimaldi[1], Goncalo Gomes[1], Alessandra Bianchi[1], Stefan Lange[2], Hylke Beck[3], Cinzia Mazzetti[4], Luc Feyen[1]

[1]European Commission, Joint Research Centre (JRC), Italy
[2]Potsdam Institute for Climate Impact Research (PIK), Member of the Leibniz Association, Potsdam, Germany
[3]King Abdullah University of Science and Technology (KAUST), Thuwal, Saudi Arabia
[4]European Centre for Medium-Range Weather Forecasts (ECMWF), Reading, United Kingdom

**Abstract**

Since 1950, anthropogenic activities have altered climate, land cover, soil properties, channel morphologies and water management in the river basins of Europe. This has resulted in significant changes in hydrological conditions. The availability of consistent estimates of river flow at global and continental level is a necessity to assess changes in the hydrological cycle. To overcome limitations posed by observations (incomplete records, inhomogeneous spatial coverage), we simulate river discharge for Europe for the period 1951 – 2020 using a state-of-the-art hydrological modelling approach. We use the new European set up of the OS LISFLOOD model, running at 1 arcminute ($\approx$1.8 km) with six-hourly time steps. The hydrological model is forced by climate reanalysis data (ERA5-land) that is bias-corrected and downscaled to the model resolution with gridded weather observations. The model also incorporates 72 surface field maps representing catchment morphology, vegetation, soil properties, land use, water demand, lakes and reservoirs. Inputs related to human activities are evolving through time to emulate societal changes. The resulting Hydrological European ReAnalysis (HERA), provides six-hourly river discharge for 282,521 river pixels with an upstream area $> 100 \text{ km}^2$. We assess its skill using 2,448 river gauging stations distributed across Europe. Overall, HERA delivers satisfying results (median KGE' = 0.55), despite a general underestimation of observed mean discharges (mean bias = -13.1%), and demonstrates the capacity to reproduce statistics of observed extreme flows. The performance of HERA increases through time and with catchment size, as well as varies in space depending on reservoir influence and model calibration. The fine spatial and temporal resolution results in an enhanced performance compared to previous hydrological reanalysis based on OS LISFLOOD for small-to-medium-scale catchments (100 - 10,000 km²). HERA is the first publicly available long-term, high-resolution hydrological reanalysis for Europe. Despite its limitations, HERA enables the analysis of hydrological dynamics related to extremes, human influences, and climate change at a continental scale while maintaining local relevance. It also creates the opportunity to study these dynamics in ungauged catchments across Europe.

# 1 Introduction

In the last century, Europe has experienced a growth in its population, economy and urbanized area (Li et al., 2021; Paprotny and Mengel, 2023). Recent decades also witnessed a rapid rise in global air temperature, attributable to anthropogenic activities (IPCC, 2023). These evolving conditions have significantly changed flows in European streams and rivers (Barker et al., 2019; Gudmundsson et al., 2021; Vicente-Serrano et al., 2019; Wang et al., 2024), leading to multiple challenges for hydrological sciences, related, for example, to long term variability, climate change, extremes or human alterations of the water cycle (Blöschl et al., 2019b). In order to assess the impacts of these changes, hydrologists need consistent, reliable and long hydrological series. Observations, despite continuous improvements (Blöschl et al., 2019a; Ekolu et al., 2022), can hamper the analysis Pan-European long-term trends due to sparse spatial distribution in some regions and temporal discontinuities. One option to overcome these limitations is to rely on a suit of models (climate, hydrological, land use) to simulate past hydrological conditions and interpret changing dynamics in the hydrological cycle in connection with rapidly changing human systems (e.g., Richards and Gutierrez-Arellano, 2022). This article introduces the Hydrological European ReAnalysis (HERA) for the period 1951-2020, providing consistent estimates of river flow for European rivers at high spatial and temporal resolution.

Hydrological models are essential tools to understand and characterise processes related to the water cycle (e.g., flood and drought forecasting). In the past three decades, there have been efforts in developing models that are able to simulate hydrological processes at large scale (continental to global scale). A myriad of these Global Hydrological Models (GHMs), differing in their conceptualization, now exist (Beck et al., 2017; Sood and Smakhtin, 2015; Kauffeldt et al., 2016; Prudhomme et al., 2011). The nature of GHMs implies that they are usually run at coarse spatial resolution (e.g., 0.5°), limiting their relevance for local and regional water resource problems (Sood and Smakhtin, 2015). Nonetheless, the development of GHMs has been fuelled by continuous improvements in remote sensing technologies and processing power (Yang et al., 2021). Remote sensing technologies provide high resolution input for hydrological models such as land use and vegetation properties. The advancements in computational capabilities have allowed to refine the spatial and temporal scale of hydrological models, enabling a more accurate representation of surface and subsurface processes and reducing modelling uncertainties (Wood et al., 2011). In this context, HERA falls within a global effort towards the development of hyper-resolution (1 km and below) land surface and hydrological models at continental (Hoch et al., 2023; O'Neill et al., 2021) and global (Hanasaki et al., 2022) scales.

A key hindrance to simulating past river flows has been the availability of meteorological inputs for hydrological models. Among potential inputs, climate reanalysis offers several advantages: temporal coverage (typically spanning several decades), a large number of variables (e.g., precipitation, wind

speed, temperature) that are physically consistent with homogeneous spatiotemporal resolution. Reanalysis data are outputs of climate models calibrated on observed data worldwide (Brönnimann et al., 2018). Here we use ERA5-land, the land component of ERA5 (Muñoz-Sabater et al., 2021). A main

advantage of ERA5-Land compared to ERA5 is its horizontal resolution, which is 9 km globally, compared to 31 km in ERA5. This enhanced resolution is obtained by downscaling meteorological variables from ERA5. The temporal resolution is hourly as in ERA5. Nonetheless, reanalysis data are obtained from short-term model forecasts and can be affected by forecast errors (Pfahl and Wernli, 2012). Variables produced in ERA5 are averages over grid cells. This averaging combined with the

relatively coarse resolution of ERA5/ERA5-land often smooths local extremes (Donat et al., 2014, Tilloy et al., 2022). To tackle this issue, we downscale and bias-correct ERA5-land with a gridded observational dataset, EMO-1 (Thiemig et al., 2022) (**Section 2.2**).

In the context of the European Flood Awareness System (EFAS), an operational system for European

flood monitoring and forecasting (https://www.efas.eu), there have been recent efforts to develop more detailed surface fields (e.g., land use, vegetation) (Choulga et al., 2023) and observational climate inputs (Thiemig et al., 2022) at a spatial resolution of 1 arcminute (1', 0.0167º, typically 1.5-3 km² over Europe). These developments come alongside improvements on the OS LISFLOOD hydrological model underpinning EFAS. OS LISFLOOD is a spatially distributed grid-based hydrological and

channel routing model which was initially developed for flood forecasting and flood risk assessment (Burek et al., 2013). However, it is also able to model effects of land use change, climate change and river regulation measures and has been used in a wide range of hydrological applications, such as mapping population under water stress in relation to how much water is reserved for the environment (Vanham et al., 2021) and projecting droughts in view of climate change (Cammalleri et al., 2020a). It

is also used in the generation of the GLOFAS-ERA5 hydrological reanalysis (Harrigan et al., 2020).

Therefore, this article brings together improvements from diverse fields (i.e., remote sensing, climate modelling, machine learning, hydrology) to generate a state-of-the-art hydrological reanalysis for a European domain that covers EU27 countries, UK, Switzerland, Iceland, Norway and the Balkan

countries (Serbia, Montenegro, Bosnia-Herzegovina, Kosovo, North Macedonia and Albania) over the past 70 years. HERA aims to reproduce as accurately as possible the evolution of the hydrological landscape of Europe by using the latest development of OS LISFLOOD (improvements in processing speed, spatial and temporal resolutions and calibration), also used in the generation of the latest EFAS v5.0 reanalysis (1991-2022) (Decremer et al., 2023) (**Section 2.1**). Climate inputs are derived from

ERA5-land, bias corrected and downscaled to 1 arcminute to improve the representation of extremes (**Section 2.2**). We generated dynamic socioeconomic inputs (water demand, land use and reservoir maps) to capture the effect of human activities on the water cycle (**Section 2.3**). These developments make this dataset the first publicly available long-term Pan-European hydrological reanalysis taking

into account the evolving socioeconomic conditions that have altered the hydrological cycle since 1951.

In **Section 3**, we assess the performance of HERA against observations from 2448 river gauges in Europe.

# 2 Method

The modelling framework developed to generate the HERA dataset is presented in a flowchart in **Figure 1**. The framework is organized around the OS LISFLOOD hydrological model that is used to simulate river discharge. For this run, we use calibrated parameters for the European setting of OS LISFLOOD developed by ECMWF in the context of the EFAS-5 calibration (CEMS-Flood online documentation, 2023). We first introduce OS LISFLOOD and its calibration procedure (**Section 2.1**). **Figure 1** also displays the main input of OS LISFLOOD: high-resolution climate inputs (**Section 2.2**), state-of-the-art static (**Section 2.3.1**) and dynamic socioeconomic maps (**Section 2.3**).

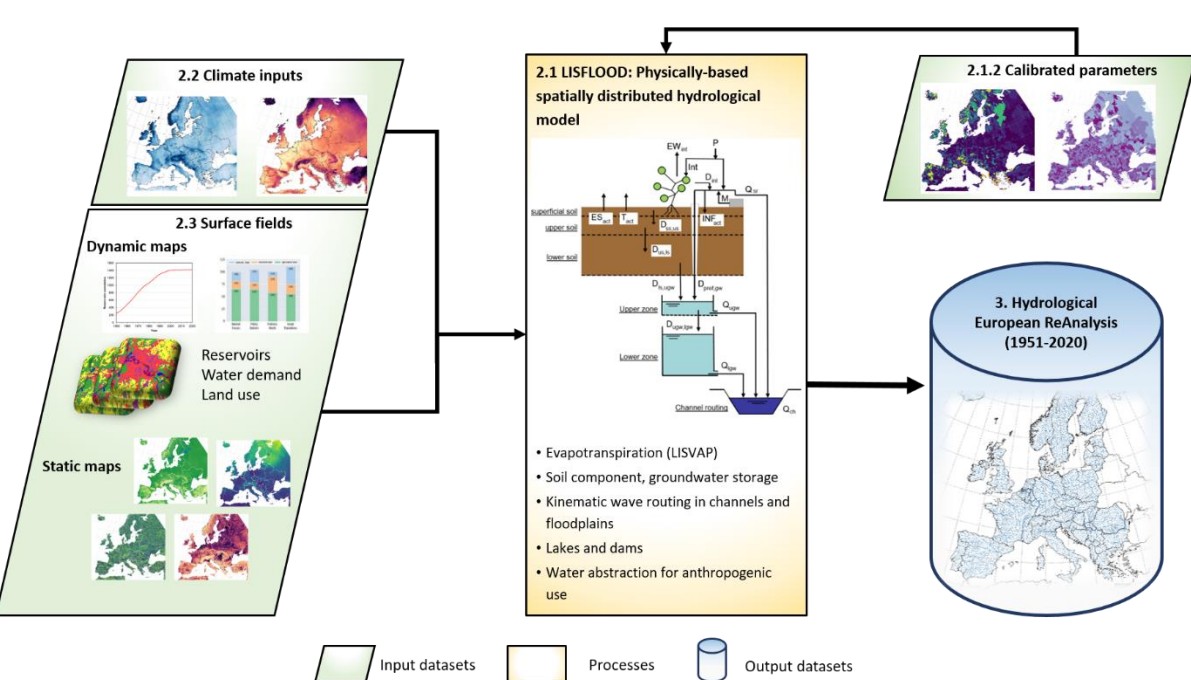

**Figure 1: Flowchart of the framework employed in the generation of HERA. Numbers relate to the section in which each component of the framework is presented.**

## 2.1  Hydrological modelling

### 2.1.1   The OS LISFLOOD model

Here, we simulate sub-daily continuous streamflow time series over Europe by means of the OS LISFLOOD model (Burek et al., 2013; Knijff et al., 2010). This is a spatially distributed, semi-physical rainfall-runoff model combined with a routing module for river channels (Dottori et al., 2022). The model has been developed by the Joint Research Centre (JRC) since the late 1990s and is used operationally for large-scale flood forecasting in the European Flood Awareness System (EFAS) and

the Global Flood Awareness System (GLOFAS). OS LISFLOOD has also been used in drought monitoring (Cammalleri et al., 2020b, 2017), to assess the effect of flood adaptation measures, environmental flow protection, or climate change (Mentaschi et al., 2020; Vanham et al., 2022). Since 2019, the model is open source and available on GitHub along with a set of auxiliary tools

(https://github.com/ec-jrc/lisflood-code). OS LISFLOOD is composed of the following main components:

- 3 soil layers (superficial, upper, lower) for water balance modelling;
- sub-models for the simulation of groundwater and subsurface flow (using 2 parallel interconnected reservoirs);

- a sub-model for the routing of surface runoff to the nearest river channel;
- a sub-model for the routing of channel flow.

Other processes such as snow melt, infiltration, rainfall interception, leaf drainage, evaporation and water uptake by vegetation, surface runoff, and exchange of soil moisture between soil layers are also simulated by the model (OS LISFLOOD online documentation, 2023). OS LISFLOOD is also able to

model lakes and reservoirs.

In this work, we use the latest version of OS LISFLOOD (v4.1.2, January 2023), which includes upgrades compared to previous versions in the hydrological routines and improvements in the modelling of water abstraction for anthropogenic use. Moreover, OS LISFLOOD v4.1.2 benefits from

improvements in the management of large inputs and in computational performance. **Figure 2** displays the domain for which data was retained in HERA. This comprises 42 European countries and excludes non-EU countries of the former Soviet Union, countries in North Africa and Middle East, and Turkey, that are included in the EFAS domain. Moreover, HERA uses the same domain as the Historical Analysis of Natural Hazards in Europe (HANZE) database (Paprotny and Mengel, 2023; Paprotny et

al., 2023). We run the model using the 1' grid used in EFAS v5.0 (Decremer et al., 2023). The temporal resolution of the simulation is 6-hourly, which is the standard for EFAS since 2020. Due to the size and spatial resolution of our domain combined with the 6-hourly time-steps, we divide the simulations in 71 yearly chunks based on calendar year starting on 3 January 1950. To estimate the initial model state, we performed a 71-years pre-run . More in particular, we used the pre-run to initialize the soil and upper

groundwater zone storages and to derive average inflow into the lower zone and discharge, which represent theoretical steady state storage. Due to the rapidly evolving socioeconomic conditions in catchments of Europe, we change the input socioeconomic maps at the start of every new calendar year of the simulation (**Section 2.4**). This differs from the standard EFAS settings, which assume static land use and reservoir network, and only varies the water demand values. At the start of every calendar year,

the model is initialized with the state variables from the last time step of the previous year (warm start). As water volumes at the first time step in the channels are not known, the model sets a conventional

initial volume (OS LISFLOOD uses half-bankful), leading to unrealistic initial discharge in some catchments. We therefore removed the first simulation year (1950) from the final dataset. Further, we only retained simulations for river pixels with an upstream area greater than 100 km$^2$, resulting in

simulations in the 282,521 river pixels displayed in **Figure 2**.

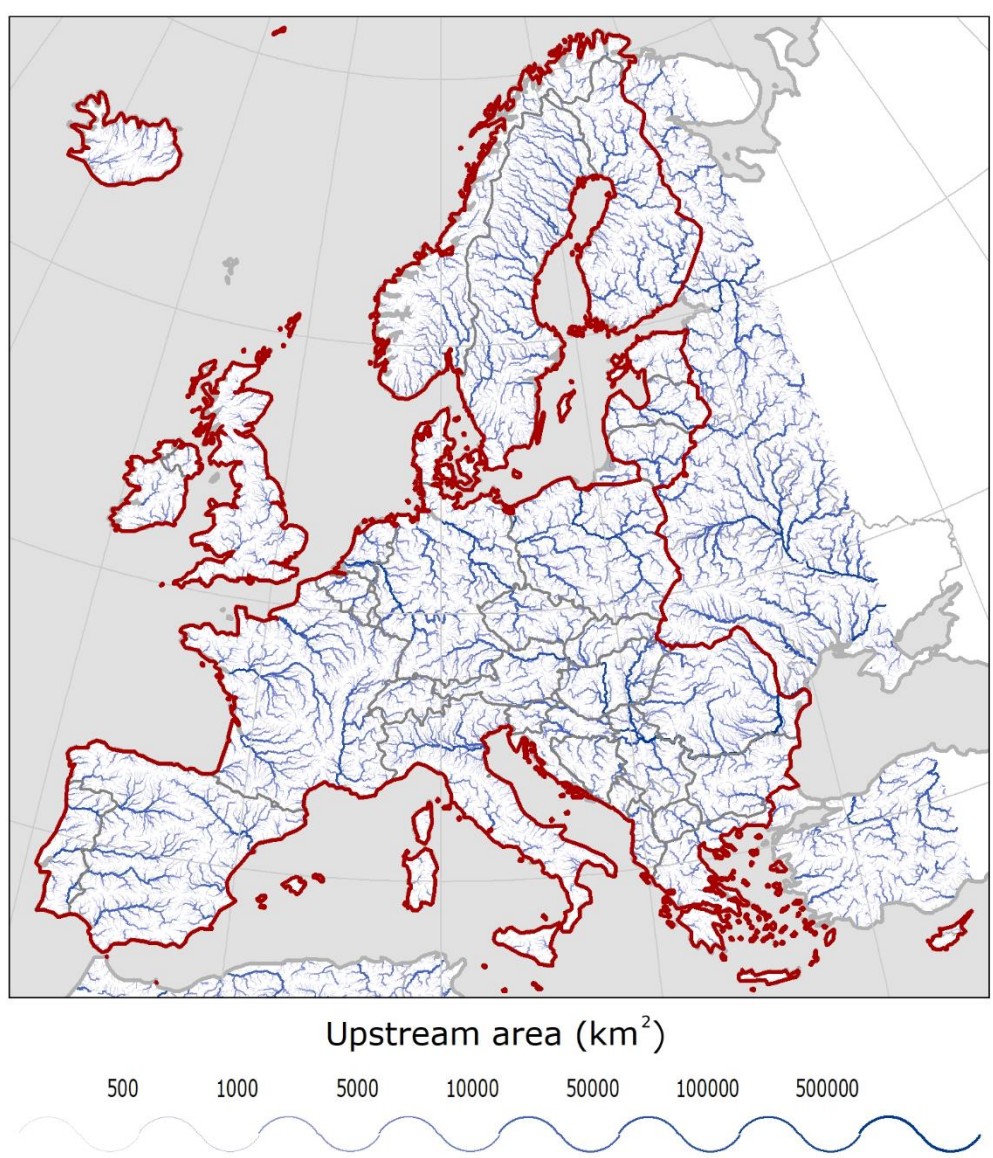

**Figure 2 River network (rivers with an upstream area > 100 km$^2$) on which discharge data has been generated. The HERA domain (in which data is provided) is confined by the red bordered area.**

### 2.1.2     Model Calibration

In this work, we also take advantage of the new EFAS v5.0 calibration that was completed in December 2022 by ECMWF. The calibration was performed using the EMO-1 meteorological dataset (Thiemig et al., 2022) over the period 1990-2021, with a focus on high flows. The modified Kling-Gupta Efficiency (KGE', Gupta et al., 2009; Kling et al., 2012) was used as a skill metric. Discharge data at 1903 stations, identified through a selection process based on several criteria (CEMS-Flood online

documentation, 2023), were used to calibrate the LISFLOOD model over Europe. Sub-daily data is always preferred when available (994 over 1903 stations). For stations where only daily observations were available, the 6 hourly discharge simulations were first aggregated to daily steps (daily mean) before evaluating the objective function. The calibration was performed at catchment level, with the 1903 selected stations covering 69.6% of the HERA domain. A map showing the calibrated catchments is provided in **Supplementary Figure S1**. The calibration was performed on 14 parameters that influence the modelling of snow melt, water infiltration into the soil, surface water flow, groundwater flow, lakes and reservoirs dynamics. A list of the calibration parameters is provided in **Supplementary Table S1**. Parameter values were identified using the Distributed Evolutionary Algorithm for Python (DEAP, Fortin et al. 2012) within a physically realistic range. The calibration protocol went from head-catchments to downstream catchments in a top-down manner, prescribing physical dependencies between upstream and downstream catchments within the same basin.

Coastal and endorheic catchments with drainage area smaller than 150 km$^2$, representing 6.5% of the HERA domain, are modelled with default parameter values. Parameter values for other ungauged catchments were estimated by parameter regionalisation. These catchments are mostly located near the coastlines, with a high concentration in southern Italy and Greece, and represent 23.9% of the HERA domain. The parameter regionalization here consists of transferring parameter values (except the ones linked to reservoirs and lakes) from a calibrated catchment to an ungauged catchment. Catchments are matched according to climatic and geographical similarities (Beck et al., 2016). We discuss the impact of calibration on the skill of HERA in **Section 3.1.1**. For more information on the calibration of EFAS v5.0, we refer to the online documentation of the Copernicus Emergency Management Service for floods (CEMS-Flood online documentation, 2023).

### 2.2    Climate inputs: Bias-adjusted climate reanalysis data

To force the hydrological model OS LISFLOOD, we used a bias-adjusted and downscaled climate dataset based on the ERA5-land climate reanalysis (Muñoz-Sabater et al., 2021). The main steps involved in the preparation of the climate inputs are summarized in **Figure 3**. The following variables are retrieved from ERA5-land at hourly temporal resolution for 1950-2020:

- Total precipitation (tp)
- Mean temperature (ta)
- Mean zonal and meridional wind speed (u, v)
- Mean dew point temperature (td)
- Total surface solar radiation downwards (ssrd)

Precipitation and temperature data were aggregated to 6-hourly resolution, and the other variables to daily resolution (**Figure 3**). All variables were averaged, except precipitation, which was summed to reach the target temporal resolution. Minimum and maximum daily temperature were also calculated, while dew point temperature was converted into relative humidity and actual vapour pressure.

Our setting of OS LISFLOOD requires meteorological data with a 1' resolution. To downscale ERA5-Land data from $0.1° = 6'$ to 1', we performed a statistical downscaling and bias adjustment using ISIMIP3BASD v3.0.0 (Lange 2019, Lange et al. 2024, Frieler et al. 2024). The ISIMIP3BASD method was initially developed for phase 3 of the Inter-Sectoral Impact Model Intercomparison Project (ISIMIP) and aims to provide robust bias adjustment of extreme values, preservation of trends across quantiles, and a clearer separation of bias adjustment and statistical downscaling compared to its predecessors (Lange, 2019). We used the new EMO-1 gridded observational dataset (1' version of EMO-5, Thiemig et al. 2022) developed for the operational EFAS-v5.0 as the high-resolution reference dataset. EMO-1 covers the period 1990−2020 and has also been used directly as climate inputs in the calibration (**Section 2.1.2**). We used 1990−2020 as the training period for the algorithm since both datasets overlap for this period. The trained algorithm is then applied to ERA5-Land to produce high-resolution data for both the training period and for 1950−1989, where high-resolution data comparable to EMO-1 are not available. The resulting climate data consistently covers 1950−2020. The ISIMIP3BASD method is applied on the following variables:

- daily mean near-surface relative humidity (hurs), obtained from actual vapor pressure (vp),
- daily and 6-hourly total precipitation (pr),
- daily total surface downwelling shortwave radiation (rsds),
- daily mean near-surface wind speed (ws),
- daily and 6-hourly mean near-surface air temperature (tas),
- diurnal near-surface air temperature range (tasrange = tasmax − tasmin),
- diurnal near-surface air temperature skewness (tasskew = (tas − tasmin)/tasrange).

Here, tasmin and tasmax are the daily near-surface air temperature minimum and maximum, respectively.

Version 3.0.0 of ISIMIP3BASD differs technically from version 2.5.0 that was used to produce the climate forcing data for phase 3b of the Inter-Sectoral Impact Model Intercomparison Project (ISIMIP3b, Frieler et al. 2024), yet both versions produce the same results, and we apply version 3.0.0 using the same climate variable-specific parameter settings as for the ISIMIP3b data production (Lange et al. 2024, Frieler et al. 2024). ISIMIP3BASD has been designed for daily data but it is applied here to bias-adjust and statistically downscale sub-daily (6-hourly pr and tas) data as if these are daily values. For the bias adjustment, a parametric trend-preserving quantile mapping method was applied to pr,

sfcwind, tas, and tasrange, while non-parametric quantile mapping was applied to hurs, rsds, and tasskew. The bias adjustment was done at the spatial resolution of ERA5-Land, 6', using spatially aggregated EMO-1 data (spatial averaging). Data resulting from the bias-adjustment were then statistically downscaled to 1' spatial resolution by using an algorithm based on the MBCn bias-adjustment method (Cannon et al., 2018) (**Figure 3**). The downscaling method is conservative in the sense that the 1' output data would be identical to the 6' input data in case the former is spatially aggregated back to 6' resolution.

Finally, potential evapotranspiration ($et_0$), potential open-water evapotranspiration ($e_0$) and potential bare soil evapotranspiration ($es_0$) are computed with bias-adjusted and downscaled data at pixel level using an approach based on the Penman-Monteith equation with the LISVAP model (LISVAP online documentation, 2023).

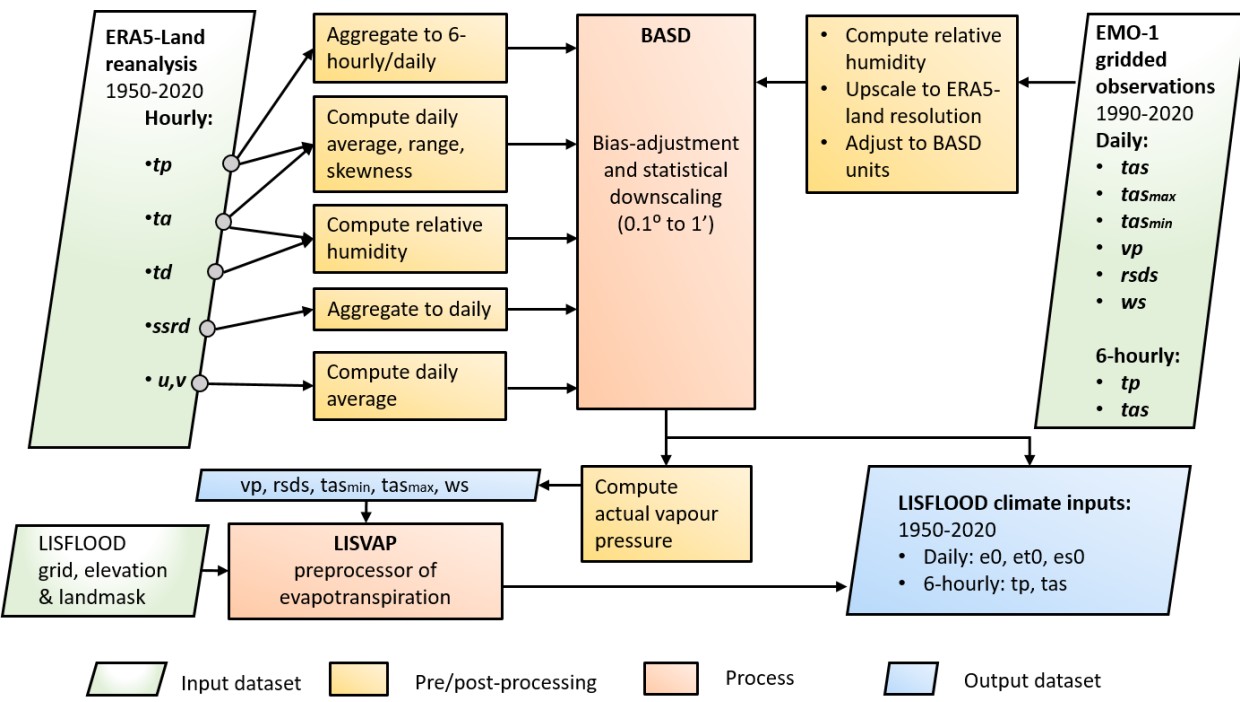

**Figure 3: Climate inputs pre-processing scheme, including temporal aggregation, bias-adjustment, statistical downscaling and processing of evapotranspiration.**

## 2.3 Surface field maps

OS LISFLOOD requires a set of surface fields maps. Depending on the model set-up it can ingest up to 108 surface fields divided in six categories:

(i)      Catchment morphology and river networks

(ii)     Vegetation cover types and properties

(iii)    Soil properties

(iv)    Land use

(v)     Water demand

(vi)     Lake and reservoir information

The first three categories, hereafter referred to as static maps, were directly taken from the
CEMS_SurfaceFields_2022 open-source dataset of the Copernicus Emergency Management Service,
developed for the European domain at 1 arc min resolution, which can be found in the JRC Data
Catalogue (Choulga et al., 2023). The last three categories were derived from
CEMS_SurfaceFields_2022 and modified to take into account socioeconomic changes (hereafter
referred to as dynamic socioeconomic maps). This section briefly presents each of the map categories,
with an emphasis on dynamic socioeconomic maps, which are original to this work.

### 2.3.1   Static maps

Static maps include surface fields of morphology and channel shapes (14 maps), vegetation properties
(18 maps) and soil properties (29 maps).

Morphology and river network information were directly used for the computation of snow melting,
temperature scaling, river routing and open water evapotranspiration. Morphologic information was
derived from elevation and includes elevation gradient, within-grid standard deviation of elevation, and
Manning's roughness coefficient. Maps representing channel shapes and river networks provide
information on grid cell area (which varies with latitude as the grid projection is WGS84), local drainage
direction, upstream area and channel dimensions. All morphology and river network maps were derived
from the Multi-Error-Removed Improved-Terrain Digital Elevation Model v.1.0.3 (MERIT DEM)
(Yamazaki et al., 2019) and the Catchment-based Macro-scale Floodplain (CaMa-Flood) Global River
Hydrodynamics Model v4.0 maps (Yamazaki, 2023).

Vegetation cover types and property maps are involved in the computation of precipitation interception,
evaporation, transpiration, surface runoff and root water uptake. These properties are described though
four variables: crop coefficients (transpiration), crop groups (water uptake), manning roughness
(surface runoff) and leaf area index (interception and evaporation). Each of these variables were mapped
for three different land cover types: forest, irrigated and other. Further, maps of planting and harvesting
days for rice, which has specific water demands, are also available. Vegetation properties were derived
from several data sources including the Copernicus Global Land Service (CGLS) Leaf Area Index (LAI)
at 1 km (Copernicus, 2021), the Spatial Production Allocation Model (SPAM) – Global Spatially-
Disaggregated Crop Production Statistics Data for 2010 (Yu et al., 2020; International Food Policy
Research Institute, 2019), and the Food and Agriculture Organisation (FAO) of the United Nations
Irrigation and Drainage Paper No.56 (Allen et al., 1998).

Soil properties refer to physical characteristics of the soil and aim to describe the water dynamics
through a vertical soil profile. In OS LISFLOOD, the soil profile is composed of three layers: superficial

(0 – 5cm), upper (5 – varying (30 – 50) cm) and lower soil layer. For each layer, variables representing soil hydraulic properties (e.g., soil moisture content, pore size index) are provided. Similarly to vegetation property maps, variables were mapped for two categories of land cover, 'forest' and 'other'. Soil properties were derived from the International Soil Reference and Information Centre (ISRIC) global gridded SoilGrids dataset (release 2017) available at 250m (Hengl et al., 2014), which is based on more than 150,000 observation sites and covariate data.

A table summarizing all the static and dynamic surface field maps used to produce HERA is provided in **Supplement Table S2**. For more details on these surface fields maps, their production and input datasets used, we refer to (Choulga et al., 2023).

### 2.3.2   Dynamic land use

OS LISFLOOD includes six land use classes as inputs: rice, other irrigated land, forest, sealed surfaces, open water, and other (non-irrigated agriculture, non-forest natural, pervious artificial); these land use classes are mostly based on CLC-Refined 2006 dataset by Batista e Silva et al. (2013) in the default setting. Among hydrological processes, interception, evapotranspiration, infiltration, and surface runoff respond differently to each land use type. With the aim to better represent complex rainfall-runoff processes, OS LISFLOOD accounts for the sub-grid variability in land use. Therefore, the spatial distribution of each land use class is defined as a percentage of the whole represented area of a given pixel (OS LISFLOOD online documentation, 2023). The magnitude of the variation of hydrological response is tied to the magnitude of the changes in land cover. De Roo et al., (2001), for instance, investigated the effects of land use changes on floods in two European catchments and identified different results depending on the magnitude of the land cover change. While such changes tend to have a limited impact on river discharge, they can locally increase flood magnitude (Merz et al., 2021; Sajikumar and Remya, 2015; Van Lanen et al., 2013; Van Loon, 2015). We modified here the grid cell fractions of each land use class using HANZE-Exposure land use maps at 100 m resolution (Paprotny and Mengel, 2023) for 42 countries in the study area. In the remaining part of the domain, we used coarser, 5' resolution maps from HYDE 3.2 (Klein Goldewijk et al., 2017) to modify the 2006 values. The temporal evolution of land area of each class is displayed in **Figure 5.a**. There has been a strong increase in sealed surfaces (+40%), while for the other relevant land use classes the changes are less than 10%, with more land occupied by irrigated agriculture (except rice), water surface (due to reservoir construction) and forests.

### 2.3.3   Dynamic water abstraction

Human water use, representing water withdrawal from the natural environment (e.g., rivers, reservoirs, groundwater) for human needs, is grouped into four main sectors: livestock, domestic, manufacturing industry, and energy production. In OS LISFLOOD, water use is supplied by surface water bodies and

groundwater depending on the sector (Choulga et al., 2023). A considerable increase in water abstraction in a region can diminish surface water resources within the same area. The model also accounts for groundwater abstraction for human use, except for flooded irrigation and cooling processes. Increased groundwater abstraction can locally reduce (or halt) baseflow. To derive monthly historic sectoral water withdrawal maps, we followed the methodology of Huang et al. (2018) and used the Food and Agriculture Organization (FAO) AQUASTAT sectoral water withdrawal data (Food and Agriculture Organisation, 2023) as a starting point. These data were subsequently spatially and temporally disaggregated using a variety of datasets. These include the Global Human Settlement Layer (Schiavina et al., 2019; Florczyk et al., 2019) for population estimates, the Global Change Analysis Model (GCAM; Calvin et al., 2019) for regional water withdrawal and electricity consumption, and the Gridded Livestock of the World (GLW; Gilbert et al., 2018) for livestock distribution. Additional datasets included the Multi-Source Weather (MSWX; Beck et al., 2022) for air temperature data, United States Geological Survey (USGS) water withdrawal estimates, and Vassolo and Döll (2005) industrial and thermoelectric withdrawal maps. More information on water demand and input datasets used is provided in Choulga et al. (2023).

We extrapolated the water withdrawal maps to the period 1950-1978 using annual gridded 0.5 degree data from ISIMIP 3a (Frieler et al., 2024; Wada et al., 2016) that were downscaled to 1' resolution using historical population data from HANZE (Paprotny and Mengel, 2023) and HYDE 3.2 (Klein Goldewijk et al., 2017) for other parts of the domain. More precisely, the ratio between EFAS high-resolution water demand maps and the ISIMIP 3a dataset for 1979 was used to adjust water withdrawal data in each grid cell. Intra-annual (monthly) cycling of water use in the energy and domestic sectors was estimated for 1950–1978 using the same approach as for 1979–2020, informed by temperature data from our input meteorological dataset (**section 2.3.1**). Livestock water use was assumed constant before 1979. Water demand and use for irrigation was computed directly by the hydrological model based on land use data and available water. The evolution of water use by sectors between 1950 and 2020 is displayed in **Figure5.c** as well as **Supplementary Table S4**. Total water use peaked in 1990 after more than doubling since the 1950s, before declining due to a drop in demand from manufacturing and energy sectors. Nonetheless, there are usually much stronger trends at country or catchment levels.

### 2.3.4   Dynamic reservoir maps

Reservoir maps contain the location and an identifier of reservoirs and are linked to tables containing metadata on storage capacity, construction year and a set of values associated to reservoir operation rules. Normal reservoir outflow rates were further adjusted through the model calibration (**Section 2.1.2**). The year of construction for each reservoir was taken from the EFAS reservoirs database, HANZE (Paprotny and Mengel, 2023), Global Reservoir and Dam Database (GRanD) v1.3 (Lehner et

al., 2011), or additional manual research for reservoirs not covered by the three datasets. The reservoir

maps are updated every simulation year (January 1$^{st}$) by adding newly built reservoirs. When a reservoir

is added, it is considered as empty and fills up according to its associated metadata. **Figure5.b** shows

the evolution of the number of reservoirs in Europe during the period 1950 – 2020. The number of

reservoirs in the model increased six-fold from 244 in 1950 to 1419 in 2020, though few were built

since the late 1980s.

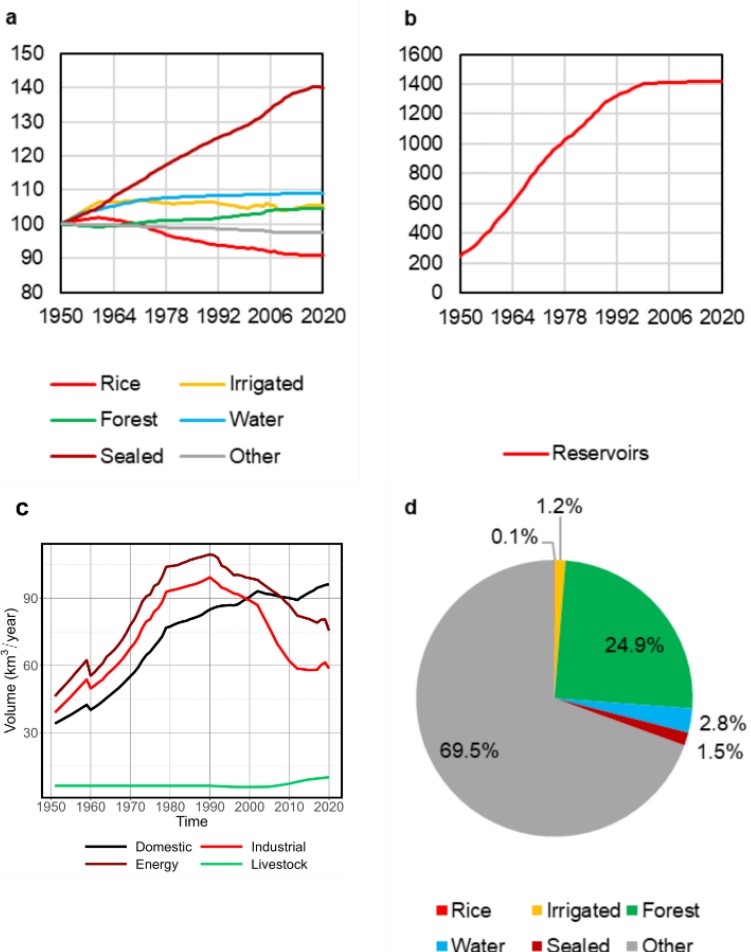

**Figure 4: Variation in socioeconomic inputs in the hydrological model, averaged over the entire EFAS domain: (a) land area by use category, 1950=100, (b) number of existing reservoirs, (c) water demand by sector in mm per grid cell per year, (d) shares of land use between the different classes in 2020.**

# 3   Results

### 3.1   Technical validation

We evaluated our hydrological reanalysis by comparison against a dataset of daily river discharge

observations from 3,442 stations across Europe. Of the data obtained, 60% were from the Global Runoff

Data Centre (GRDC) and 40% from national public datasets of France, Norway, Poland, Spain, Sweden

and the United Kingdom. Furthermore, this dataset was compiled independently from the one used in

the EFAS calibration (**Section 2.1.2**). The stations' record duration varies between 1 and 71 years. The selection of stations used for validation is based on several criteria:

- Spatial matching: To link stations to their corresponding river pixel, we scanned the nine modelled pixels around the river gauge location. When information on the upstream area was available (for 60% of the stations), we retained the pixel with the closest upstream area to the reported one. For pixels without information on the upstream area, we retained the one with the closest simulated mean discharge ($Q_{mean}$) to the observed one. For a more accurate spatial matching, we used the available LISFLOOD coordinates from the EFAS calibration (1026 stations). A total of 546 stations did not match with LISFLOOD river pixels, mostly due to their upstream area being lower than 100 km$^2$.

- Upstream area verification: The spatial matching selected the closest upstream area for stations where we have information on catchment area. It is however possible that the reported catchment differs largely from its matched pixel upstream area. We removed stations where the difference between the pixel and observed upstream area was larger than 50% (51 stations).

- Mean discharge comparison: For some stations, the ratio between observed and simulated $Q_{mean}$ was suspicious. This could be due to an erroneous spatial match (i.e., matching of a river with a station on a tributary). As uncertainty grows with smaller streams, we decided to remove those with a suspicious $Q_{mean}$ ratio ( $r_{Qmean} > 6$ or $r_{Qmean} > 3$ if $Q_{mean,obs} > 10$ m$^3$/s) (49 stations)

- Manual check: A manual verification was performed on 66 stations with KGE'<-0.41. Each station and its matching pixel were individually checked, resulting in the removal of 13 more stations due to wrong spatial matching, erroneous station location, and doubtful observations. The corresponding river pixel was manually set for 8 stations. Manually checked stations and the reason for their exclusion/inclusion are provided in **Supplementary Table S5**.

- Finally, we removed stations with a record length shorter than 30 years (334 stations). This enabled a meaningful comparison between different locations in the validation process.

This procedure resulted in the selection of 2,448 river stations across Europe, with an upstream area ranging from 100 to 785,421 km$^2$. Among these stations, more than half (1,507) have an upstream area of less than 1000 km$^2$ and a fifth (498) have an upstream area of less than 200 km$^2$.

The HERA reanalysis comes at a sub-daily resolution (6-hourly), but the performance could only be evaluated at the daily time step of the observational dataset. Discharge data from HERA was therefore aggregated (daily mean) for the technical validation. We expect performance to be slightly higher at daily scale, as the temporal aggregation tends to increase the correlation between observed and modelled discharge. Performance was assessed using the KGE' on discharge data (Gupta et al., 2009; Kling et al., 2012). KGE' was used as the standard performance metric in EFAS and GLOFAS (Harrigan et al.,

2020; Cammalleri et al., 2020b), as well as in other hydrological model assessments (Lin et al., 2019; Harrigan et al., 2020; Beck et al., 2017) and is composed of three components: correlation, bias errors, and variability errors:

$$KGE' = 1 - \sqrt{(r-1)^2 + (\beta - 1)^2 + (\gamma - 1)^2} \qquad (1)$$

$$\beta = \frac{\mu_s}{\mu_o} \qquad (2)$$

$$\gamma = \frac{\sigma_s / \mu_s}{\sigma_o / \mu_o} \qquad (3)$$

where $r$ is the Pearson correlation coefficient between simulated (s) and observed (o) flow, $\beta$ is the bias ratio, $\gamma$ is the variability ratio, $\mu$ the mean discharge, and $\sigma$ the discharge standard deviation. KGE' and its three components are dimensionless with an optimal value on 1. It is important to note here that KGE' values should not be interpreted like the more traditional Nash-Sutcliff efficiency (NSE, Nash and Sutcliffe, 1970). Indeed, for KGE' the mean flow benchmark has a value of $KGE' = 1 - \sqrt{2} = -0.41$. Any value above -0.41 therefore exceeds the benchmark (Knoben et al., 2019), meaning that the model performs better than simply taking the mean.

In **Section 3.1.1**, we assessed model performance across space, time (1951-2020) and catchment size, in order to identify strengths and weaknesses of HERA. Despite covering many aspects of the performance of hydrological models, KGE' mainly focuses on mean values and give a higher weight to high extremes compared to low ones. As this dataset also aims to be used for long term analysis of hydrological extremes, we also evaluated how well high and low extremes are reproduced, including their timing and seasonality.

### 3.1.1 Hydrological performance

We quantified here the overall performance of HERA in terms of KGE' as well as the decomposition of this indicator into its three components: correlation, bias and variability. **Figure 5** displays the distribution of KGE' and its three components across the 2,448 validation stations. We obtained a KGE'>-0.41 for 2,411 (98.5%) of them, meaning the reanalysis is skilful for these stations (**Figure 5.a**). The median KGE' across all catchments is 0.55 while the mean is 0.46, although this value varies widely across catchments (**Figure 5.a**, **Figure 6.a**). The mean correlation value is relatively high ($\bar{r}$= 0.69) with 90% of the stations having $r$>0.5 (**Figure 5.b**). From **Figure 5.c** and **Figure 5.d**, we can observe that there is a tendency to slightly underestimate flows ($\bar{\beta}$= -13.1 %) and flow variability ($\bar{\gamma}$= -14.2%). The bias ranges between $0.8 - 1.2$ ($0.5 - 1.5$) in 50% (91%) of the river gauges, which is considered as very good for hydrological reanalysis (Harrigan et al., 2020; Alfieri et al., 2020; Lin et al., 2019; Yang et al., 2021).

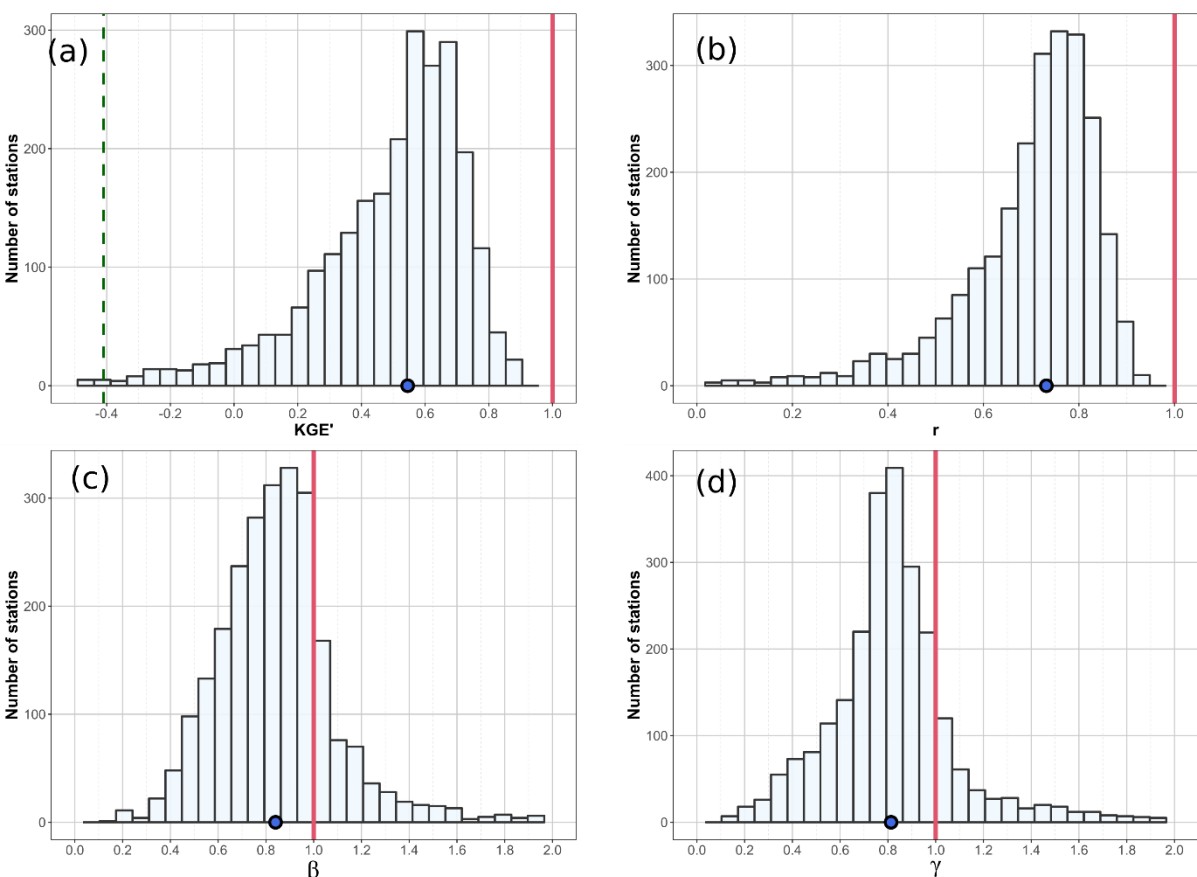

Figure 5: HERA hydrological skill for the 2,448 selected stations in terms of (a) KGE' and its three components: (b) Pearson correlation, (c) bias ratio, (d) variability ratio. In (a), the green dashed vertical line represent the benchmark KGE' value ('-0.41). The red vertical line represents the ideal values and the blue dot represents the median for all stations.

**Figure 6** shows the spatial performance of the model in terms on KGE' and its components. The highest skill can be observed in central and north-western Europe. The vast majority of stations in UK, Germany, France, Austria, Switzerland (which together account for 51% of all 2,448 stations) exhibit a good (>0.5) to very good (>0.75) KGE'. On the other hand, performance is relatively poor in Spain, Cyprus, Scandinavia and Northern Poland. Factors that can explain the poor performances in southern Europe include the combination of arid climates and the strong influence of lakes and reservoirs (**Figure 7.c**). Dry catchments where precipitation events are separated by long dry spells are in general very difficult to model (Cantoni et al., 2022). In Scandinavia, the negative bias (**Figure 6.c**) could be linked to an underestimation of precipitation and snowmelt in Scandinavian mountains (Beck et al., 2017, 2020). **Figure 6.d** presents the variability ratio of simulated to observed flow. Overall, our reanalysis exhibits lower variability than observations, with 83% of the catchments having a variability ratio below one. The underestimation of variability was also found in the EFAS v5.0 run, although it is more pronounced in HERA. This could be explained by the different meteorological forcing used in the two runs.

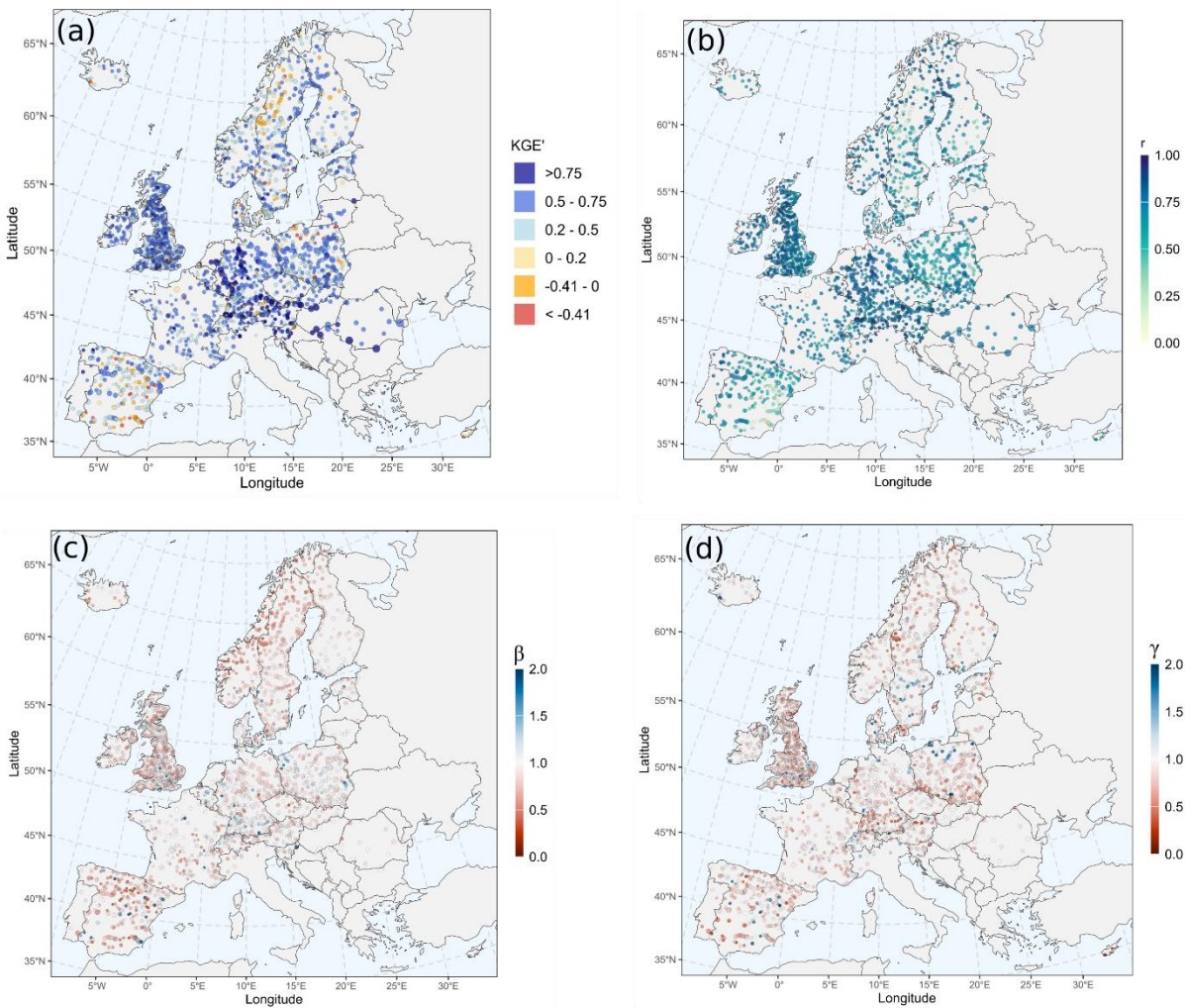

**Figure 6: KGE' and its three components: (b) Pearson correlation, (c), bias ratio, (d) variability ratio at the 2448 river gauges considered in the validation of HERA. Point size are proportional to catchment size.**

We validate HERA on stations with a wide range of catchment area (mean upstream area of 7,615 km$^2$), which has an impact on OS LISFLOOD performance (Harrigan et al., 2020). The set of 2,448 validation stations includes stations that were used in the calibration process (596) as well as stations in uncalibrated catchments (36) (See **Supplementary Figure S1**). In **Figure 7**, we break down the performance of the reanalysis according to different attributes of each catchment: time (**Figure 7.a**), catchment area (**Figure 7.b**), reservoir impact (**Figure 7.c**) and calibration status (**Figure 7.d**).

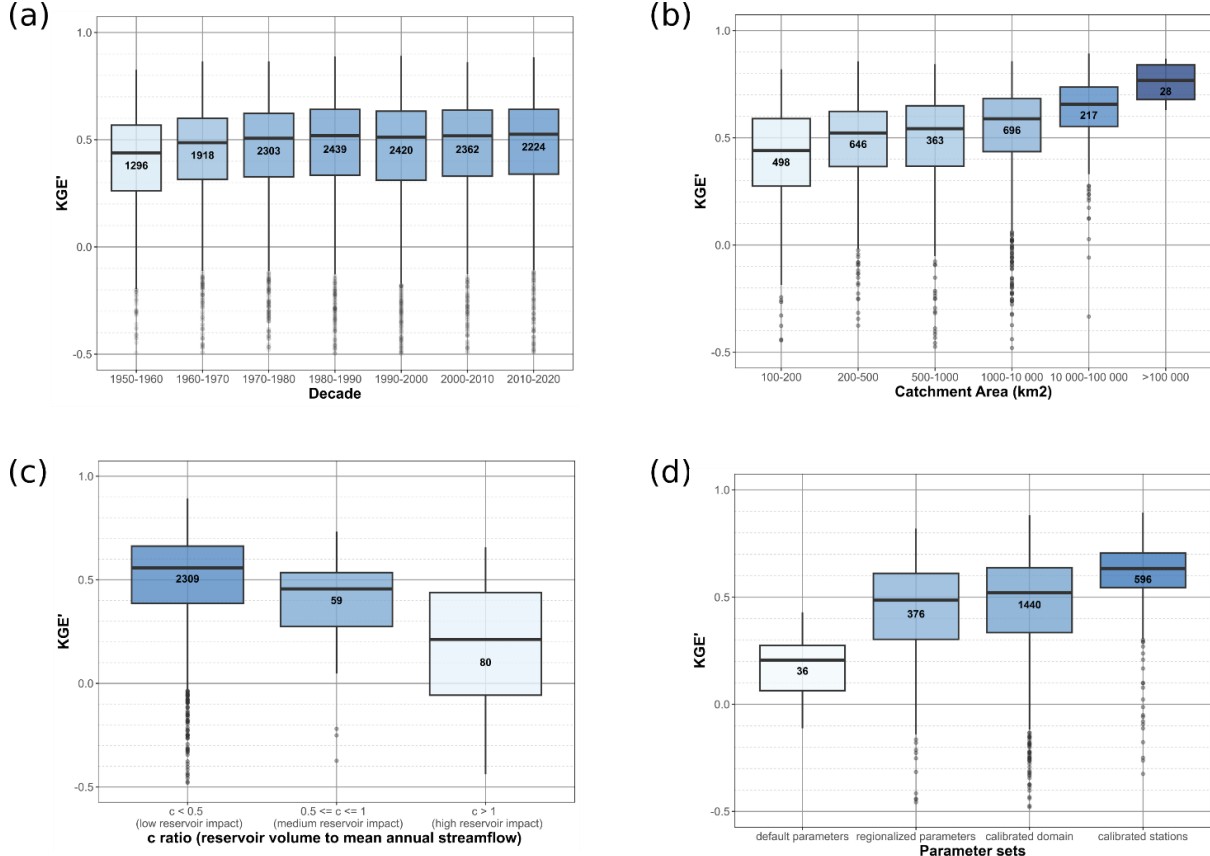

**Figure 7: Boxplot of HERA KGE' according to different classifications of the 2,448 river stations used in the validation, (a) time, (b) catchment area and (c) reservoir impacts. Numbers inside boxplot represent the amount of river gauges for each category, while the colour of the boxplot represent the median performance of the group from low (light blue) to high (dark blue).**

Overall, the skill of HERA shows a slight increase through time with an increase of 21% of the KGE'$_{med}$ between the 1950s in the 2010s. The skill increases between 1951 and 1980 and then stabilizes from 1981 to 2020, though the results are influenced by changes in gauge data availability over time. It could also be driven by improved climate inputs. **Figure 7.b** shows that model skill increases with catchment size, from a KGE'$_{med}$ of 0.44 (IQR 0.25 – 0.59) for the 498 smallest catchments (<200 km$^2$) to 0.77 (IQR 0.68 – 0.84) for the 28 largest catchments (>100,000 km$^2$). Such patterns have already been observed at global scales (Harrigan et al., 2020). It is important to note here that the majority of stations used in this validation (62%) have an upstream area below 1000 km$^2$ and the median upstream area of the 2448 stations is 583 km$^2$. This is half of the median upstream area of the 1903 stations used in the calibration of EFAS-5 (CEMS-Flood online documentation, 2023).

We also divided stations according to reservoir influence. From the 1420 reservoirs active in 2020 (which represent the maximum amount over the considered time window), we estimated the impact of reservoirs on streamflow at grid-cell level. This was done by computing the ratio (c [-]) of reservoir volume to mean discharge (Nilsson et al., 2005) at every grid cell. The ratio has been computed with the accuflux function from PCRaster and compares the upstream cumulative reservoir capacity [m$^3$] and the cell-specific annual volume of annual streamflow [m$^3$] (Zajac et al., 2017). This ratio varies

between 0 and 1608 downstream of Embalse de Finisterre in central Spain. Most of the river grid-cells highly impacted by reservoirs are found in southern Europe, particularly in Spain and Bulgaria. **Figure 7.c** highlights the influence of reservoirs on the skill of the reanalysis. River cells affected (medium and high, c >0.5) only represent 6% of stations and grid cells in the domain (**Figure 2**). Median skill is the lowest for highly impacted (c > 1) stations, with $KGE'_{med} = 0.24$, whereas minimally impacted stations have a $KGE'_{med}$ of 0.55. This highlights the difficulty of large-scale hydrological models such as OS LISFLOOD to accurately simulate reservoir outflows (Zajac et al., 2017).

Finally, we investigated the influence of calibration on the model skill. In **Figure 7.d**, River gauges are divided into four groups according to their calibration status. As displayed in **Supplementary Figure S1**, 83% of the stations considered in the validation fall into the domain calibrated for EFAS v5.0. We find a better performance for calibrated stations, ($KGE'_{med} = 0.64$) and a comparable skill for stations within the calibrated domain ($KGE'_{med} = 0.52$) and stations benefitting from the parameter regionalization ($KGE'_{med} = 0.47$). The performance is much lower for catchments with default parameters, which here are limited to small (< 150 km$^2$) coastal and endorheic catchments.

### 3.1.2 Reproduction of extremes

Large scale hydrological models forced by climate reanalysis often fail to reproduce extreme hydrological event characteristics in part due to the coarse spatial and temporal resolution (Brunner et al., 2021b; McClean et al., 2023). Here, we analyse how well HERA reproduces different flow quantiles (q05, median, q95) through the Person correlation coefficient and the coefficient of determination ($R^2$) (**Figure 8**) for the 2448 considered catchments. The ability to capture annual maxima/minima and their seasonality is also assessed (**Figure 9**).

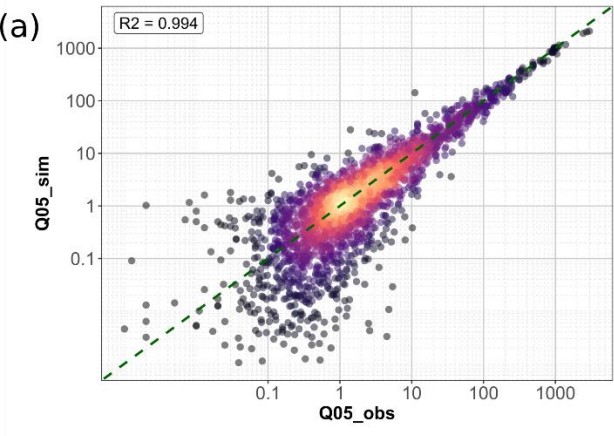

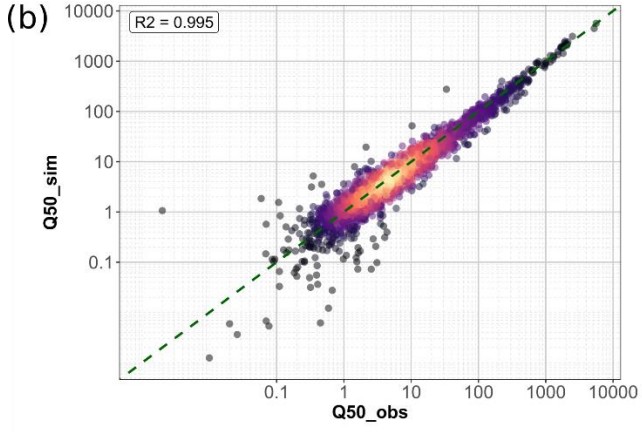

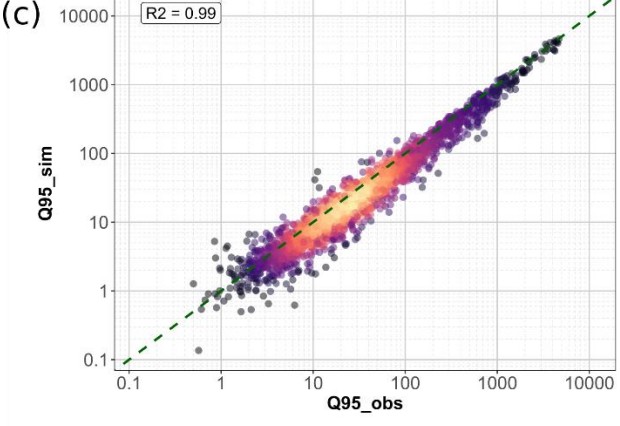

**Figure 8: Scatterplot of observed versus simulated river flow quantiles [m³/s]: (a) 5% quantile, (b) median (q50), (c) 95% quantile (q95) for the 2448 River gauges.**

**Figure 8** displays scatter plots of observed versus simulated quantiles. Each point represents one of the 2448 stations. We observe that low (5% quantile: $Q_{05}$) and median ($Q_{50}$) flows are generally well represented with $R^2 > 0.99$ (**Figure 8.a** and **Figure 8.b**), especially for larger discharge values. However, despite this generally good agreement, there is a more pronounced deviation of simulated values from observations for lower flow values, expressed by a higher dispersion for $Q_{05}$. These deviations can be attributed to bias in climate inputs (McClean et al., 2023), the hydrological model

(Feyen and Dankers, 2009), but also to errors in flow measurements, especially for $Q_{05}$ (Despax, 2016; Tomkins, 2014) and anthropogenic impacts on low and median flow regimes (Brunner, 2021a) that are not accurately represented in the model (see **Figure 8.c**). The number of stations with large deviations in the reproduction of high flow statistics ($Q_{95}$) is minor compared to $Q_{05}$ and $Q_{50}$. Nonetheless, despite a relatively high $R^2$ (0.99), there is a general underestimation in the simulated values (**Figure 8.c**), which is common for large scale hydrological models. Similarly to low and median flows, errors in high flow statistics can be due to biases and smoothing of extremes in climate inputs and errors in the hydrological modelling. Uncertainty associated to flow measurements also play a major role for high flows, as rivers discharge are usually not directly measured during floods (Despax, 2016). Finally, the spatial and temporal resolution of the model can affect its ability to reproduce high flows, particularly for flash floods in small catchments.

We also assessed the ability of the reanalysis to reproduce the timing of annual maxima and minima of discharge as well as their overall seasonality. As the daily temporal scale is not the most relevant when it comes to drought analysis with discharge data (Hannaford and Marsh, 2006; Kohn et al., 2019), annual minima were computed from 30-day moving average flows. **Figure 9.a** displays the mismatch in mean day of occurrence computed with circular statistics following Berghuijs et al. (2019). We observe that the median error in the mean day is very close to zero for both maxima (median = 0.1, IQR=-12 – 18) and minima (median= -1, IQR = -28 – 41), but with a much higher dispersion for annual minima compared to the annual maxima. The higher dispersion for low flows is due to the slow-onset nature of these events (Brunner et al., 2021a). **Figure 9.b** shows the difference in timing between simulated and observed annual maxima across the 2448 considered stations. Differences in timing are smaller over the Atlantic coast though a particularly high lag (simulated maxima delayed by 30 days or more in HERA) is observed over Poland and central Spain. For low flows (**Figure 9.c**), delays in central Europe are larger than 30 days, while in Scandinavia the timing can be up to several months too early. This can be explained by the high hybridity of river regimes (several high and low flow seasons) in these regions, which may be captured with varying accuracy in HERA.

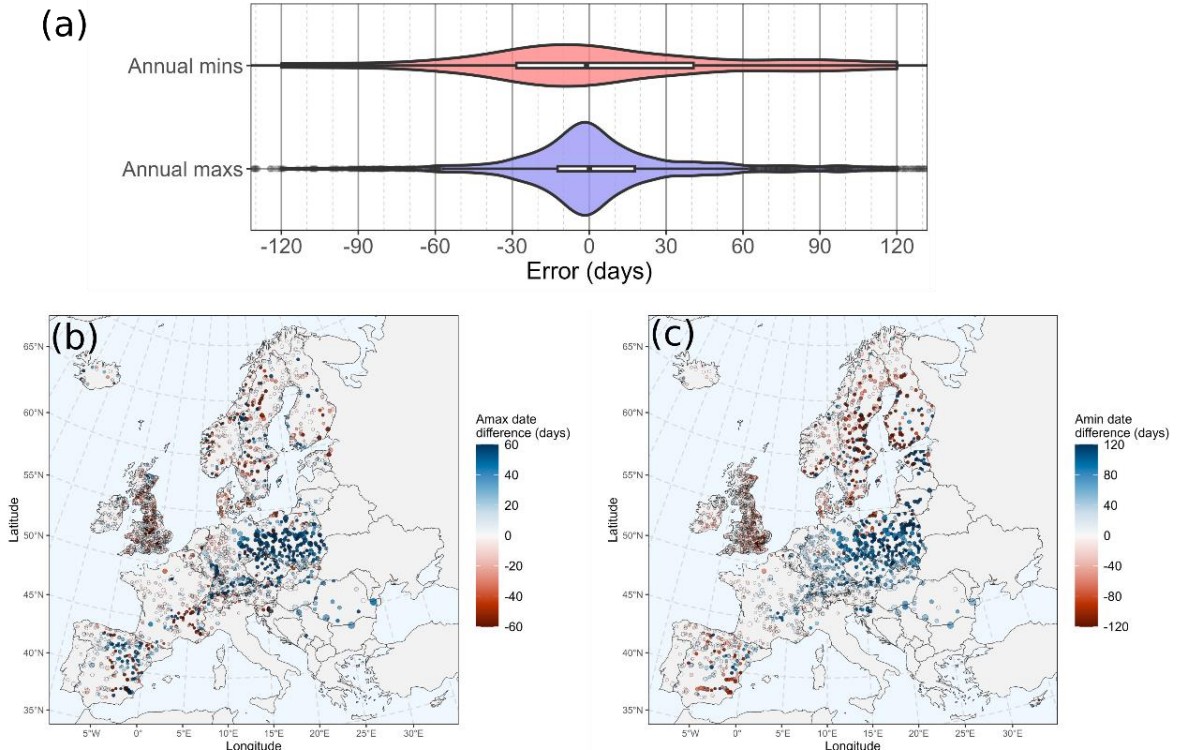

**Figure 9: Assessment of the ability of HERA to reproduce the timing of annual maximum and minim flows. (a) Violin plot of error in mean day of occurrence of annual maxima (daily discharge) and minima (30-day averaged discharge) computed with circular statistics. Inside each violin plot, boxplots display the median, 1st and 3rd quantiles. (b) Difference between the modelled and observed mean annual maxima date (positive value means a later occurrence in HERA). (c) Difference between the modelled and observed mean annual minima date (positive value means a later occurrence in HERA)**

In addition to the validation protocol presented in this section, we compared reported performances of HERA with other recent hydrological datasets and carried out a comparison between HERA and another recent hydrological simulation done with the grid-based conceptual mesoscale Hydrological Model (mHM) (Kumar et al., 2013; Samaniego et al., 2010; Samaniego et al., 2019, Thober et al., 2019) for Europe for the period 1960-2010. More details on the comparison are provided in Supplementary material (**Figure S3-S6**).

### 3.2 Usage notes

HERA brings together several improvements (climate, scale, socio-economic dynamics) to better simulate river discharge in catchments of Europe over the past 70 years. Despite covering still a relatively short period of time compared to human history on earth, these 70 years capture a very intense period of climate and socioeconomic change, often called the Anthropocene, and offers multiple research opportunities:

- Assessment of long-term trends in European river regimes

- Provide benchmark data for "data poor" areas

- Generate catalogues of flood and drought events

- Identification of spatial and temporal correlations between European catchments

- Identification of changes in hydrological extremes characteristics (frequency, magnitude, timing)

- Combination with other data products for compound hazard analysis

- Provide scenarios for flood inundation simulations

In this section, we briefly present a possible usage of the data, addressing changes in regime for diverse rivers across Europe (**Figure 10**).

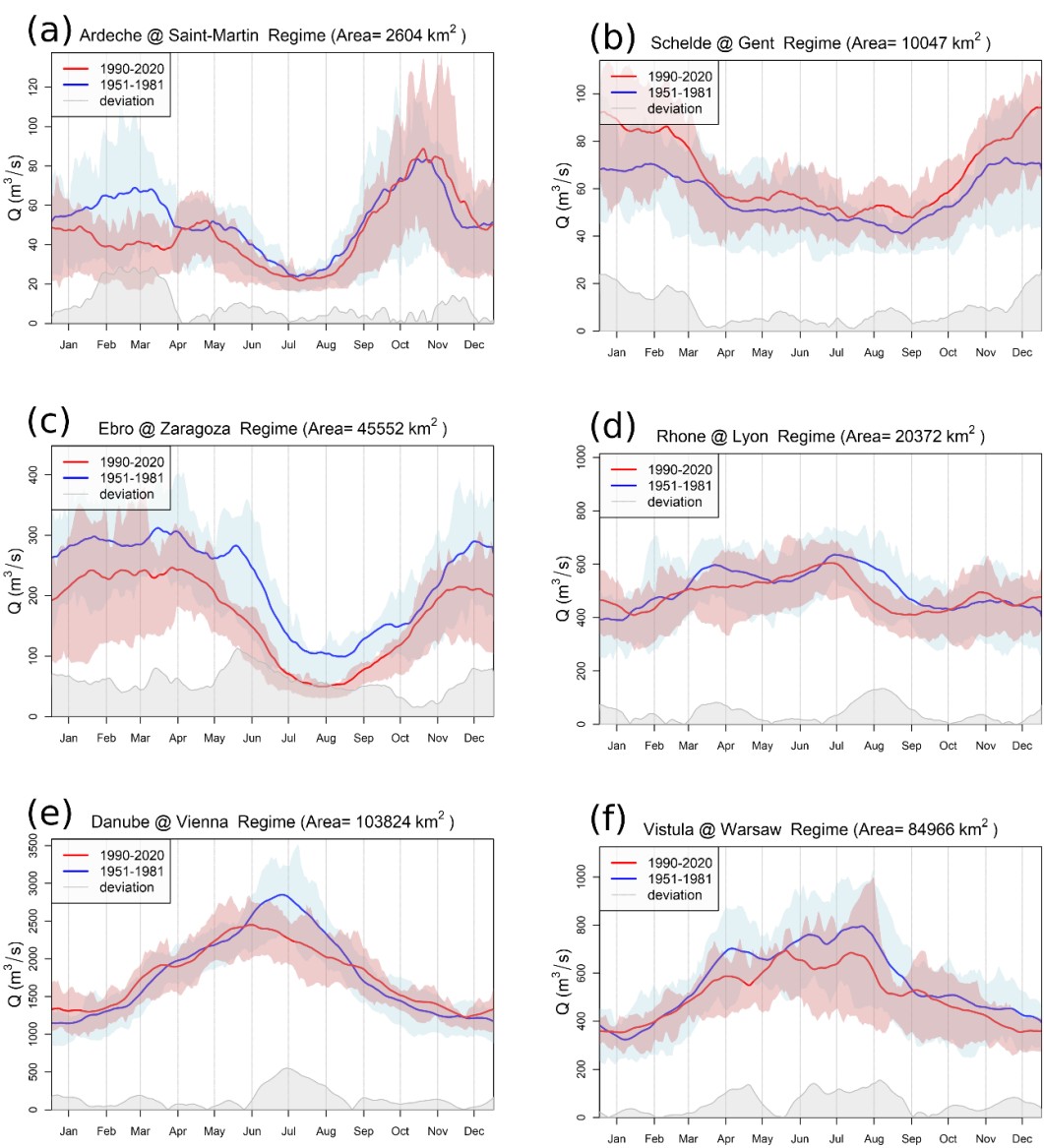

**Figure 10: Changes in flow regime between 1951-1981 (blue) and 1990-2020 (red) for six diverse European rivers: (a) Ardèche, (b) Schelde, (c) Ebro, (d) Rhone, (e) Danube and (f) Vistula. The regime is computed here as the 30-day moving average. Shaded coloured areas represent the IQR of discharge for every day of the year. The grey shaded area represents the absolute difference between the two regimes corresponding to different periods.**

**Figure 10** displays hydrological regimes, here represented as the mean of a 30 day's average moving window over a given period, for six European rivers. These rivers differs in terms of hydrological regimes, with three main regimes represented:

- Mediterranean pluvial regime for the Ardèche (a), with its recognisable high flows in autumn.
- Pluvial or oceanic regime for the Schelde in Ghent (b) and the Ebro in Zaragoza (c)
- Nival regime for the upper Rhone in Lyon (d), the Danube in Vienna (e) and the Vistula in Warsaw (f).

These six rivers also vary in terms catchment area, geographic location (France, Austria, Poland, Belgium, Spain), climate (Mediterranean, Continental, Oceanic, Alpine) and geomorphological conditions. For each river the flow regime for 1951 - 1981 (in blue, first 30 years of HERA) and 1990 – 2020 (in red, last 30 years of HERA) are shown. By comparing the two regimes, one can observe diverging patterns of changes among these rivers. For the two pluvial rivers, the Schelde and the Ebro (**Figure 10.b-c**), both pluvial rivers, we observe opposite patterns of change, the Schelde saw an increase of its average discharge throughout the year while the Ebro experienced a downward shift in regime. For the upper Rhone and Danube (**Figure 10.d-e**), which are influenced by snowmelt in their upper catchments, we see lower and earlier flow peaks in spring and summer. The Vistula (**Figure 10.f**) saw an overall increase in flow throughout the year. Finally, the Ardèche (**Figure 10.a**) has seen reduced flow throughout the year with a notable decrease in late winter which can be associated to the reduction of snowfall in the Massif Central where the Ardèche has its up waters (François et al., 2023). The timing of the autumn peak seem to have slightly shifted towards earlier dates, in agreement with a recent study on trends in Mediterranean floods (Tramblay et al., 2023).

# 4 Discussion

Recent developments in diverse fields, including climate, hydrology, remote sensing and computational sciences, have made the generation of high-resolution reanalysis products possible (Aerts et al., 2022; Hanasaki et al., 2022; Hoch et al., 2023). In this context, HERA brings discharge data for all European rivers with upstream area larger than 100 km$^2$ for the period 1951-2020. With its refined spatial and temporal resolution, HERA represents hydrological processes in Europe with more detail than previous publicly available hydrological reanalysis products (Harrigan et al., 2020; Schellekens et al., 2017). Calibrating hydrological models can significantly improve river flow simulation (Beck et al., 2017; Kauffeldt et al., 2016). Parameters in 93.5% of the HERA domain were adjusted during a calibration process (**Section 2.1.2**) or parameter regionalization (Beck et al., 2016). This is a very high calibration coverage for a GHM (Beck et al., 2017), that can be explained by the relatively high coverage in river gauging stations in Europe.

It is difficult to compare HERA with other recent hydrological reanalyses such as GLOFAS-ERA5 (Harrigan et al., 2020) and GRFR (Yang et al., 2021), for several reasons: (i) spatial coverage (global vs continental), (ii) spatial resolution (0.25º, 0.05º, 0.0167º), (iii) temporal coverage (iv) dynamic vs static socioeconomic conditions. We provide however a short summary of reported performances of HERA, GLOFAS-ERA5, GRFR and a European-scale hydrological simulation with the mHM model (EUmHM) in **Supplementary Table S6**. While the reported performances of HERA are higher than its global counterparts, thse are very close to the performance of EUmHM. In a more detailed comparison with EUmHM over 515 European river gauges (see **Supplementary Material Figure S3-S6),** we show that HERA generally outperforms the EUmHM run in terms of KGE' (**Figure S4**) but both models exhibit strengths and weaknesses spatially **(Figure S5**) and in terms of the components of KGE' (**Figure S6**). Differences in performances between the HERA and EUmHM run can be attributed to the many different features in the two runs, such as meteorological forcing, resolution, calibration, and flow routing within the hydrological model. Conversely, HERA shares a great number of features with the EFAS v5.0 reanalysis (Decremer et al., 2023), with slightly lower performance (not shown here). Nonetheless, EFAS v5.0 only covers the period 1990 – 2022 and assumes static socioeconomic conditions (land use, water abstraction, reservoirs).

Similarly to other aforementioned hydrological reanalyses, HERA exhibits reduced performance in cold and semi-arid catchments. This can be related to deficiencies in the representation of snow processes within OS LISFLOOD or the underestimation of precipitation at northern latitudes (Beck et al., 2017, 2020). Semi-arid environments are notoriously challenging areas for hydrological models due to the highly non-linear rainfall-runoff response and lower precipitation data quality (Cantoni et al., 2022). GHMs tend to poorly represent runoff in small-to-medium size catchments (10-10,000 $km^2$) (Harrigan et al., 2020; Sood and Smakhtin, 2015), and nearly 90% of the catchments used in the validation of HERA (**Section 3.1**) are small-to-medium size catchments. The drop in performance with smaller catchment area in HERA remains, however, moderate compared to the GLOFAS-ERA5 global hydrological reanalysis (Harrigan et al., 2020). The presence of reservoirs also influences the performance of reanalyses. While including reservoirs in the hydrological modelling has a positive impact on model performance (Zajac et al., 2017), there is still a high level of uncertainty regarding the operating rules of each reservoir. Moreover, the 1422 reservoirs used to generate HERA most likely represent just a fraction, mainly the largest ones, of all operational reservoirs in the modelled domain (Speckhann et al., 2021). In summary, the main strength of HERA lies in its relatively low bias in comparison to the other hydrological datasets considered here (**Table S6**, **Figure S6**), while its performances are hampered by its underestimation of variability.

HERA is generated through hydrological modelling, which brings a suite of uncertainties that can be divided into four categories: (i) model inputs, (ii) model structure, (iii) parameter values and (iv) observations. It remains challenging to quantify these uncertainties, however, the quality of inputs, and more in particular climate inputs is often referred to as an important factor of uncertainty (Beck et al., 2017; Sood and Smakhtin, 2015). Despite efforts in bias correction and downscaling of the climate input, it seems that on average, HERA slightly underestimates river discharges, with a more pronounced bias for high flows. As reported in other studies, negative biases can be related to an underestimation of precipitation in the climate inputs, in particular for extreme events (McClean et al., 2023; Mahto and Mishra, 2019), in high latitudes and in (semi-)arid catchments (Beck et al., 2016; Sood and Smakhtin, 2015, Hirpa et al., 2018). Model structure can also play an important role, as shown in **Figure S6**, where EUmHM is the best model in terms of correlation while HERA exhibits smaller bias ratio. This can be the result of different choices made in the main equation behind the two models, resulting in different responses to forcings and calibration. The large impact of model selection on streamflow and trend estimates is now increasingly acknowledged (Karlsson et al., 2016; Clark et al., 2016). Calibration generally improves streamflow simulations (Hirpa et al., 2018) and HERA also shows a better performance for stations used in the calibration process (**Figure 7.d**). The negative biases and variability ratios can be related to the different meteorological forcing (EMO-1) used in the calibration, although an underestimation of the variability was also found in the EFAS v5.0 run (that is forced by EMO-1). The method, parameters and skill metrics used for calibration further affects the uncertainties. Despite its qualities, the skill metric used for the calibration presented in **Section 2.1.2** (KGE') is known to result in an underestimation of variability (Brunner et al., 2021b) and to put more weight on high values (Garcia et al., 2017). This could partly explain the reduced performances in reproducing extreme low flows observed in **Figure 8** and **Figure 9**. Other uncertainties can arise from surface field maps (**Section 2.3**) and measuring of river discharges (instruments and rating curves). With sparser gauging and more complex hydraulic conditions for high and low flows, uncertainty rises (Despax, 2016).

# 5 Data availability

The HERA hydrological reanalysis and its climate and dynamic socioeconomic inputs are available via the JRC data catalogue (doi 10.2905/a605a675-9444-4017-8b34-d66be5b18c95). **Table 1** provides a brief description of the dataset and **Table 2** gives a general overview of the content of the dataset.

**Table 1: Description of the HERA dataset**

| DATASET DESCRIPTION | |
|---|---|
| Data type | Gridded |
| Projection | WGS 1984 – EPSG 4326 |
| Spatial coverage | EU27, UK, Switzerland, Iceland, Norway, Serbia, Montenegro, Bosnia-Herzegovina, Kosovo, North Macedonia, Albania |
| Temporal coverage | 01-01-1951 to 31-12-2020 |
| Temporal resolution | Six-hourly data |
| File format | netcdf |

The dataset consists of three distinct folders that are described here and in **Table 2**:

- Climate inputs: folder containing the climate forcing for the LISFLOOD hydrological model. Out of the five variables provided, three are at daily temporal resolution, potential evapotranspiration, potential evaporation and potential evaporation from bare soil (obtain with LISVAP, LISVAP online documentation, 2023), while two have a six-hourly time step,
precipitation and temperature. The spatial resolution of the climate inputs is 1'. The files are in netcdf format with one file per year per variable for a total of 355 files (2.3 TB of data).

- Socioeconomic inputs: folder containing the dynamic surface fields maps (**Section 2.3**), divided into three categories: land use, reservoirs and water demand. The land use subfolder contains 426 yearly files (4.6 GB) of land use fraction maps for each six land use classes. The reservoir
subfolder hosts 71 yearly files (3.6 GB) of reservoir location and identifier. Reservoirs are added/discarded from the simulation every year according to their construction/destruction data. Finally, the water demand subfolders contain four files (3.9 GB) representing water demand for the considered sectors (**Section 2.3.3**). Each file contains monthly maps of water abstraction for a given sector. All socioeconomic inputs are provided in the netcdf format.

- River discharge: this folder contains river discharge netcdf files for each year at six-hourly time step for all European rivers with an upstream area greater than 100 km$^2$ (2.3 GB per file, 166 GB total).

All data share the same projection (WGS 84) grid and spatial resolution (1'). Static surface fields maps were directly retrieved from the OS LISFLOOD static and parameter maps for Europe (2024) dataset, which were developed in the context of the new EFAS deployment (Decremer et al., 2023). It is
important to note that HERA simulates discharge on a slightly smaller domain than the original EFAS domain, the mask used for HERA is also provided in the dataset.

**Table 2: List of inputs and outputs of LISFLOOD provided in the HERA database (link here).**

| Subfolder | File | Resolutions | Variable/content | Unit |
|---|---|---|---|---|
| | area_hera_01min.nc | 1' | *mask of the hera domain* | |
| climate_inputs/ e0 | e0_yyyy.nc | 1', daily | *potential evaporation computed with lisvap from downscaled and bias-corrected actual vapour pressure, solar radiations, min/max daily temperature and 10m wind speed.com* | mm.d$^{-1}$ |
| climate_inputs/ et0 | et0_yyyy.nc | 1', daily | *potential evapotranspiration computed with lisvap from downscaled and bias-corrected actual vapour pressure, solar radiations, min/max daily temperature and 10m wind speed.com* | mm.d$^{-1}$ |
| climate_inputs/ es0 | es_yyyy.nc | 1', daily | *potential evaporation from bare soil computed with lisvap from downscaled and bias-corrected actual vapour pressure, solar radiations, min/max daily temperature and 10m wind speed.* | mm.d$^{-1}$ |
| climate_inputs/ pr6 | pr6_yyyy.nc | 1', six-hourly | *downscaled and bias-corrected six-hourly precipitation* | mm.d$^{-1}$ |
| climate_inputs/ tp6 | ta6_yyyy.nc | 1', six-hourly | *downscaled and bias-corrected six-hourly average temperature* | °c |
| socioeconomic_ maps/landuse | fracforest_european _01min_yyyy.nc | 1', yearly | *fraction of pixel area covered by evergreen and deciduous needle leaf and broad leaf tree areas* | |
| socioeconomic_ maps/landuse | fracsealed_europea n_01min_yyyy.nc | 1', yearly | *fraction of pixel area covered by urban areas, characterizing the human impact on the environment* | |
| socioeconomic_ maps/landuse | fracirrigated_europ ean_01min_yyyy.nc | 1', yearly | *fraction of pixel area covered by irrigated areas of all possible crops excluding rice* | |
| socioeconomic_ maps/landuse | fracwater_european _01min_yyyy.nc | 1', yearly | *fraction of pixel area covered by rivers, freshwater and saline lakes, ponds and other permanent water bodies over the continents* | |
| socioeconomic_ maps/landuse | fracrice_european_ 01min_yyyy.nc | 1', yearly | *fraction of pixel area covered by irrigated areas of rice* | |
| socioeconomic_ maps/landuse | fracother_european _01min_yyyy | 1', yearly | *fraction of pixel area covered by agricultural areas, non-forested natural area, pervious surface of urban areas* | |
| socioeconomic_ maps/reservoirs | res_european_01mi n_yyyy.nc | 1', yearly | *location and identifier of each reservoir* | |

| | | | | |
|---|---|---|---|---|
| socioeconomic_maps/water_demand | dom_1950_2020.nc | 1', monthly | *daily supply of water volume for indoor and outdoor household purposes and for all the uses that are connected to the municipal system (e.g., water used by shops, schools, and public buildings)* | mm.d$^{-1}$ |
| socioeconomic_maps/water_demand | ene_1950_2020.nc | 1', monthly | *daily supply of water volume for fabricating, processing, washing and sanitation, cooling or transporting a product, incorporating water into a product* | mm.d$^{-1}$ |
| socioeconomic_maps/water_demand | ind_1950_2020.nc | 1', monthly | *daily supply of water volume for the cooling of thermoelectric and nuclear power plant* | mm.d$^{-1}$ |
| socioeconomic_maps/water_demand | liv_1950_2020.nc | 1', monthly | *daily supply of water volume for domestic animal need* | mm.d$^{-1}$ |
| river_discharge | dis.herayyyy.nc | 1', six-hourly | *river discharge for river pixels with upstream area>100km$^2$.* | m$^3$.s$^{-1}$ |

# 6 Conclusion

Despite the limitations discussed above, HERA represents a state-of-the-art, high-resolution, long-term hydrological reanalysis for Europe in the form of homogeneous river flow data generated with the OS LISFLOOD model. To our knowledge, no other publicly available hydrological reanalysis currently provides discharge data at similar scales and spatiotemporal coverage for Europe. The inclusion of dynamic socioeconomic conditions provides a more realistic reanalysis of river flows in heavily managed European catchments. The increased spatial resolution improves the performance due to a better representation of hydrological processes and inputs required to simulate them, including the river network (Hoch et al., 2023; Thober et al., 2019). HERA advances the reanalysis of extreme hydrological events, notably by the sub-daily temporal resolution and high-resolution bias corrected climate input. The magnitude and seasonality of extremes are fairly reproduced, even if biases exist in some regions (e.g., central Poland, southern Spain). The dataset covers 70 years and is therefore suited for the analysis of long-term trends of several hydrological signatures. The modelling framework developed here further forms a basis for creating alternative (counterfactual) time series of river discharges where climatic or socioeconomic conditions can be kept static, enabling the attribution of changes in hydrological regimes across Europe (Kreibich et al., 2019; Sauer et al., 2021; Scussolini et al., 2023).

## Supplement

### Author contribution

**Aloïs Tilloy**: conceptualization, data curation, formal analysis, software, writing – original draft preparation. **Dominik Paprotny**: conceptualization, methodology, formal analysis, writing – original draft preparation. **Stefania Grimaldi**: methodology, software, supervision. **Cinzia Mazzetti**: methodology, software. **Goncalo Gomes**: software. **Alessandra Bianchi**: visualization. **Stefan Lange**: conceptualization, methodology, writing – reviewing and editing. **Hylke Beck**: conceptualization, methodology, writing – reviewing and editing. **Luc Feyen**: conceptualization, methodology, supervision, writing – reviewing and editing.

### Competing interests

The authors declare that they have no conflict of interest.

### Acknowledgments

The authors want to thank Larisa Tarasova, Oldrich Rakovec and Rahini Kumar for kindly agreeing to share outputs of their mHM European run, enabling a comparison with HERA. We also want to thank Pratik Mishra for proof reading the manuscript. Dominik Paprotny was supported by the German Research Foundation (DFG) through project "Decomposition of flood losses by environmental and economic drivers" (FloodDrivers), grant no. 449175973.

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
