# Peer review of "HERA: a high-resolution pan-European hydrological reanalysis (1951-2020)"

_Earth System Science Data, 2024_

## Author Response (AR1)

**REVIEWER 1**

**Short summary:**

The authors present a high-resolution model-based runoff reanalysis for the period 1950-2020 which is based on bias-adjusted and downscaled ERA5-land meteorological inputs. In the paper the general framework and incorporated datasets have been described, as well as a model evaluation of mean and extreme statistics performed. Within this framework anthropogenic influences (e.g., water use, land use change) have been considered for the period 1950-2020 and many reservoirs have been included in the hydrological reanalysis.

**General statements:**

While the development of high-quality hydrological reanalysis products is relevant and I acknowledge the large effort in creating this dataset, I have some general comments on the paper as well as some detailed comments below which I believe will help future users of the dataset. Firstly, I have noticed that several in text citations are not included in the reference list of the manuscript. Please check all your references for correct referencing in the text and the reference list. I have not checked all references but noticed numerous inconsistencies, some examples highlighted in the comments below. Further, the manuscript needs another careful reading to eliminate many typos and grammatical errors. I have initially highlighted a few errors, but stopped because there are too many. Also, the figure referencing throughout the manuscript can be improved. In some parts (highlighted below) your methodological descriptions could benefit from a few more details.

From a methodological perspective I have a few comments related to your model evaluation. I didn`t fully understand your grid cell matching approach for the comparison of LISTFLOOD with gauging stations. Maybe you could elaborate on this more and see my detailed comment on this below. Further, for your evaluation it is not clear on which time period your comparison between observations and LISTFLOOD runoff is based on. It sounds like each catchment is based on any period from 1 to 71 years of data. This needs to be clarified to better interpret your entire model evaluation. With a variable evaluation length your runoff quantiles will not be comparable across catchments. Lastly, your final data product has a temporal resolution of 6-hours, however, your entire analysis is based on daily data. Hence, we can't assess whether the performance of the reanalysis remains the same between daily and 6-hourly resolution. Some of your evaluation metrics might change. Therefore, at some point in your manuscript or in the Supplementary material you should show some evaluation of the 6-hourly data in comparison to the daily scale to showcase that your analysis is valid also for the higher resolution final product.

**General answer:**

We want to thank Anonymous Referee #1 for providing valuable and detailed comments, which will certainly improve the quality of the manuscript. Regarding the issues with the reference list, grammar and typos, we will work on these for the revised manuscript. We will also rewrite the explanation of the grid cell matching method to improve clarity.

Thank you for highlighting the issue with variable period for the evaluation of the model. In the revised version, we will filter stations based on the length of available observations (at least 30 or 40 years), this should enable a more robust comparison between catchments.

Regarding the temporal resolution, I agree with your concerns. Unfortunately, the limitation lays here on the observational data side, which we only have at daily scale. As stated in one of our answers below, the calibration done before this run (for EFAS v5) included both daily and sub-daily data. Authors of this study do not have access to the data used for the calibration of EFAS v5. However, we do not expect

a massive change of performances between daily and sub-daily data. Most likely, we would notice a small degradation in the correlation between observed and simulated data.

Also, the model we use is called LISFLOOD and not LISTFLOOD.

**Detailed comments:**

P3-L86: "environmental" -> "environment"

R: change will be implemented.

P4-L107: "fist" -> "first"

R: change will be implemented.

P4-L116: If "Here" stands at the beginning of the sentence a comma "," is needed after "Here,".

R: change will be implemented.

P4-L118: Dottori et al. 2022 is not in the reference list.

R: reference will be added.

P5-L145: "1`" Are you using a regular grid?

R: yes, each pixel is 1' x 1'.

P5-L147: Maybe clarify here that your yearly chunking is based on the calendar year. The reader can assume this from line 149 but it would be good to clarify this here already.

R: we will clarify this.

P5-L148: "71-year pre-run". What is your 71-year pre-run? Please briefly explain.

R: This sentence explaining the role of the pre-run will be added to the revised manuscript: "To estimate the initial model state, we performed following a 71-years pre-run. More in particular, we use the pre-run to initialize the soil and upper groundwater zone storages and to derive average inflow into the lower zone and discharge, which represent theoretical steady state storage."

P6-L149: Please briefly explain how you initialize the start of the new calendar year. I assume by the storage components of the previous simulation year. Also, how do you initialize when you for example introduce new reservoirs.

R: Yes, we perform "warm starts" using as initial state the state variables from the last time step of the previous year. For socioeconomic maps, each year has corresponding land use maps and reservoir maps. When a new reservoir is added, it is at the beginning of a calendar year (construction year), and the reservoir is initially set as empty. We will add a sentence explaining this in revised the manuscript.

P6-L159: "data discharge" -> "discharge data"

R: change will be implemented.

P6-7: Model Calibration section. Can you elaborate a bit more on the calibration of the model. For example, on (a) whether the model was calibrated towards floods or more the general water balance, (b) whether the model was calibrated on daily or sub-daily timescales. It is difficult to find the relevant information in the online documentation that you cite.

R: In order to calibrate LISFLOOD, 1903 sub-catchments were used, each of which is closed by a station that has at least four equivalent years' worth of discharge observations within the period from 1990 to 2021. To answer you two specific questions:

(a) The calibration used in this reanalysis was initially performed in the context of the European flood awareness system (EFAS). LISFLOOD was therefore calibrated primarily towards flood, with KGE' used as a skill metric. The parameter values of these catchments were identified using the Distributed Evolutionary Algorithm for Python (DEAP, Fortin et al. 2012). DEAP was used to explore the parameter space and identify the parameter set leading to the highest value of the modified Kling Gupta Efficiency (KGE', Gupta et al., 2009).

(b) EFAS calibration is performed using both 6-hourly and daily calibration stations. Although sub-daily data is always preferred when available, some stations used for EFAS calibration only have daily data (909 over 1903). EFAS uses a dual calibration approach: LISFLOOD runs at 6-hourly steps, then at daily stations, discharge simulations are aggregated to daily steps before evaluating the objective function.

We will add a short paragraph summarizing the information provided here in Section 2.1.2.

P7-L179: What does "modified version" mean in this context? Does it mean, bias-adjusted and downscaled or is it related to something else?

R: By modified version, we mean indeed bias corrected and downscaled. We will clarify that sentence in the revised manuscript.

P7-L188-89: Can you please clarify how you aggregated the hourly to 6-hourly/daily data. Aggregated is ambiguous and can mean taking the average, sum or instantaneous value.

R: All variables were averaged, except precipitation, which was summed to reach the target temporal resolution. We will clarify this in the revision.

P8-L198-99: Can you provide a reference for your statement of "robust bias adjustment of extremes". In the Lange 2019 citation that I believe you are referring to, I can`t find sufficient evidence to back your statement. Is this the Lange 2019 publication (https://gmd.copernicus.org/articles/12/3055/2019/gmd-12-3055-2019.html) you are citing?

R: Yes it the same reference. The sentence I am referring to can be found the abstract and is the following:" *In comparison to their predecessors, the new methods allow for a more robust bias adjustment of extreme values, preserve trends more accurately across quantiles, and facilitate a clearer separation of bias adjustment and statistical downscaling".*

P8-L220-22: Can you please elaborate on 6-hour bias adjustment modification more. It is not clear how you bias correct the 6-hour data. Do you define correction factors for the four 6-hour timesteps per month?

R: We split the 6-hourly temperature data by time of the day, which can be 0, 6, 12, or 18 UTC. This way, we obtain four subsets of temperature data (e.g., the first subset contains all data points at 0 UTC). We then bias-adjust each of those subsets independently, as if they contained daily data, which the ISIMIP3BASD method is designed to be applied to. The union of the bias-adjusted subsets then

constitutes the bias-adjusted 6-hourly temperature data. The same procedure is applied to the 6-hourly precipitation data. We will add a sentence summarizing this answer in the revised manuscript.

P8-L225: How was the EMO1-data aggregated?

R: The data was aggregated by averaging it spatially. We will specify this in the revised manuscript.

P8-L225-227: Can you elaborate a bit more on the statistical downscaling. "Statistical relationship" is a bit vague. This description seems to deviate from the ISIMIP method.

R: The method is the same as in the cited paper (Lange 2019), though indeed "statistical relationship" is not a good description of it. In a nutshell, the ISIMIP3BASD model does statistical downscaling using a bias adjustment approach. First, coarse-resolution data are interpolated to the fine-resolution grid. Then, multivariate quantile mapping is applied to adjust the multivariate distribution of all interpolated time series of climate data per coarse-resolution grid cell, effectively adding the spatial variability that is missing in the interpolated. This algorithm is based on the MBCn bias adjustment method by Cannon (2017).

P12-L323: How did you extrapolate the water withdrawal? Linearly?

R: The data was not extrapolated linearly, but proportionally to the values in the alternative dataset (ISIMIP 3a water withdrawal downscaled by population distribution). It could be alternatively explained that the annual time series of the alternative dataset for 1950-78 in each grid cell were proportionally adjusted according to the ratio between EFAS high-resolution water demand map and the alternative dataset for 1979 in each grid cell. We will specify the extrapolation method in the revised manuscript.

P14-L358: Your criterion of station selection is not based on the record length. In the previous sentence you however mention that you have records ranging from 1 to 71 years, so does this mean, that for some stations your performance metric is based on 1 year only and others on 71 years of data? Can you please clarify this in the text. If this is the case, you might need to flag this in a way or check your performance results for subsets of stations. All stations vs. at least 30-year of stations vs. only a few years of data. This is otherwise an uneven performance quantification.

R: We agree with your comment. As mentioned in the reply to the general comment, the large disparities in record length bias our performance quantification. We will redo this latter for the revised manuscript only retaining stations with at least 30 or 40 years of record.

P13-14 L359-363: Spatial matching. How is the matching of river pixels and gauging stations done in the calibration? Is your approach different from this?

In EFAS, all stations are mapped on the 1 arcmin LISFLOOD drainage network using geographical coordinates, metadata (such as drained upstream area, river name, and basin name), and, where available, information from the previous EFAS 4.2 (such as whether the location of the station was moved upstream or downstream of a confluence). The process involves matching each station with a model grid cell using geographic coordinates. Then, stations with a mismatch in the drained area between LISFLOOD and stations metadata are shifted upstream-downstream by a few grid cells; finally, stations to grid cell matching is manually inspected, and expert judgement is used to shift stations where metadata is missing. Errors in the stations to grid cell matching that can be linked to a significant mismatch between the station's upstream drained area and the upstream drained area of the matching grid cell are identified through simulated river discharge when possible.

In the validation of HERA, we used a similar approach (i.e. matching each station with a model grid cell using geographic coordinates). However, we did not have information on upstream area at river

gauges; we then used average discharge as a criterion for selecting the most likely matching grid cell for every station.

"LISTFLOOD coordinates", does this mean that you use the same river pixels from the EFAS calibration? Also, not all of your 9 grid cells will be river pixels, right? If I am not mistaken LISTFLOOD has a routing scheme so for your comparison with a gauging station, you should likely use data from the routed river pixel. Wouldn`t it be more straightforward to find the closest river pixel and base the matching on the 2-3 upstream and 2-3 downstream river pixels?

R: Exactly. Even if we use a different set of river stations than the one used for the EFAS calibration, the two sets have stations in common. For these 1026 stations, we use the grid coordinates from the lisflood calibration. For the remaining stations, we use the spatial and mean discharge match described above. The problem with the spatial matching is that for some stations, the reported location or the LISFLOOD routing scheme can be slightly inaccurate, resulting in a station not matching with any river pixel. We believe that the approach you are suggesting would produce very similar outcomes to the one performed by the authors.

P14-L379: "Performance at daily scale". Why daily scales and not 6-hourly scales? Would you expect that there is a noticeable difference in the performance metric when evaluated on a daily or 6-hourly scale?

R: In this study, the main reason behind the usage of daily scale is the temporal resolution of observations, which are daily.

EFAS calibration is performed using both 6-hourly and daily calibration stations. Although sub-daily data is always preferred when available, some stations used for EFAS calibration only have daily data (909 over 1903). EFAS uses a dual calibration approach: LISFLOOD runs at 6-hourly steps, then at daily stations, discharge simulations are aggregated to daily steps before evaluating the objective function.

We do not expect a noticeable degradation in the performance metric for 6-hourly calibrated stations when evaluated on a daily scale. The temporal aggregation tends to smooth the simulated hydrograph, reducing the variability between simulated and observed discharge and increasing the correlation.

P14-L388-89: KGE of -0.41. This is not clear. Clarify that you mean with this that you consider KGE values between -0.41 < kge < 1 as reasonable values and that your simulation is better than simply taking the mean values. You can use your sentence from P15-L403: "[…] a KGE>-0.41, meaning the reanalysis is skillful for these stations."

R: change will be implemented.

Figure 5: Potentially the median is more meaningful than the mean here. In the figure caption the description of the dashed line is missing. Further, the ECDF is not explained. In the text you are also not commenting on the ECDF, therefore I would suggest removing the ECDF from the panel (a).

R: Figure will be edited accordingly.

P16-L420: "Baltic countries" What is your hypothesis for the Baltic countries? Also, snowmelt processes? I briefly checked your Figure S2 and noticed that for the Baltic countries you have only a short record length. Therefore, I wonder if the poor performance in the Baltic regions is not related to the short record length. This directly corresponds to my comment from above on the inconsistencies with regard of the record length.

R: Thank you for this comment. Indeed the record length has a higher influence on the KGE' than I thought initially. For the revised manuscript, I will look into relationship between record length and performances, especially for Baltics countries.

P16-L421: "southern Europe". Do other studies, based on LISTFLOOD, also show this poor performance in southern Europe? Could this be a general issue of LISTFLOOD for dry catchments? Within the Alps many rivers are heavily influenced by reservoirs, and we don't see the degradation in performance. It could be worth checking other LISTFLOOD studies for similar results and potential explanations.

R: Many possible causes can explain degraded LISFLOOD performance in Southern Europe. In Spain, there is a large number of reservoirs, and the model struggles to represent reservoir outflows; in Albania, observed discharge time series were short and sometimes of low quality; in Israel, LISFLOOD struggles to represent sparse events. Dry catchments exhibiting precipitation events between long dry spells are, in general, very difficult to model with any hydrological model. Such difficulties in arid catchments have also been observed in a recent study (Cantoni et al., 2022) for Tunisa. We will add a few sentence to this paragraph to clarify our explanation.

P16-L424: What is "negative 1.2es"?

R: It is bias, no idea how it became 1.2es…We will correct this.

P16-L426: "lower variability". Is this a general issue of LISTFLOOD or is this due to lower variability in ERA5-land?

R: According to performance assessment of the two latest hydrological datasets generated with LISFLOOD (GLOFAS v4 and EFAS v5 long runs), it seems that a lower variability is a general issue with LISFLOOD. However, the issue is more pronounced in HERA compared to EFAS v5. One explanation could be that the calibrated parameters from EFAS v5 were computed with the EMO-1 gridded observational dataset and not ERA5-land bias corrected. We also consider more (2900 vs 1900) and smaller (minimum area=100km2 vs 150) that EFAS v5, which could explain some of the performance degradation. We will add one sentence to the paragraph summarizing this answer.

P18-L443: Sometimes you write HERA and other times HER. Please check for consistency.

R: It is HERA, we will remove all instance of HER.

P19-L473: Grammar. Here, we analyse how well ...

R: change will be implemented.

P19-L474: How long is your observational record for this analysis? In a previous section, and mentioned in a previous comment, you state that your records are between 1 and 71 years long. However, I have never seen a time constraint for the record length. Therefore, I am wondering here whether for some stations your quantiles are based on only 2 years of data while others represent 60+ years of data. Please clarify this. If this is the case, then your flow quantiles among the different stations mean totally different things.

R: Indeed, each station has a different length of record. As stated in previous replies, a minimum record length will be added to ensure compatibility between results from different stations.

P19-L474: "Person" -> "Pearson"

R: change will be implemented.

P21-L505: I think elaborating on the spatial patterns of the differences in timing would be worth including. For the mins 50% of your catchments show biases larger than 25 days. Figure 9 could be extended with two Maps, (eg. a) Map of differences in Annual-mins, b) Map of differences in Annual-max, c) violin plots.

R: The spatial patterns are indeed relevant here: We will add the maps to Figure 9 as suggested and elaborate on the latter and potential explanations.

P21-L506: The annual maxima/minima are not necessarily a flood or a drought event. I would rephrase this to "annual high and low flows".

R: change will be implemented.

Figure 10: Labels are quite small, especially the subplot titles. Please add (a)-(f) and include the rivers in the figure caption (e.g., (a) Ardeche, …). The ordering of the regime examples is also random. It would be easier to order them by regime type. Further, please extend the figure caption. What is the shading? Is the solid line the 30-year average?

R: The figure will be modified accordingly.

P24-L540f: Please include appropriate figure referencing in the text (e.g., Figure 10b, c). Is connected to the above comment on Figure 10.

R: We will include appropriate referencing in the reviewed manuscript.

P24-L540-542: "For instance, …" I would suggest rephrasing this sentence. For example: "For the two pluvial rivers, the Schelde and the Ebro (Figure reference), we see opposite patterns of change. The Schelde shows an increase […], while the Ebro […]."

R: We will rephrase the sentence in the revised manuscript.

P24-L542: In the same sentence as above. Is this really a water deficit? Meaning it is not enough water? I would rather suggest to phrase this as "the entire regime of the Ebro shifts downward throughout the entire year."

R: I agree that this is rather a shift than a deficit. The sentence will be changed.

P24-L567-68: Can you briefly comment on how sensitive the LISTFLOOD model is to land use and water use changes? In most hydrological models very large disturbances or changes in the land use must be present to even see any influence on the hydrological response. Land use changes are very small as you have shown in your descriptions and these changes are based on the entire domain, so I would not assume any major differences arising from static and dynamic land use changes. Accounting for water use and its changes could locally certainly have some implications.

R: Thank you for this question. In OS LISFLOOD, land use affects the following processes: evapotranspiration, canopy interception, infiltration and runoff, water storage on surface depressions. Soil properties (hydraulic conductivity, residual and saturated soil moisture content, Van Genuchten parameters) directly depend on land cover, to adequately account for differences between hydrological processes in forests, and areas with low vegetation.

LISFLOOD has been designed to account for differences in hydrological processes driven by different land cover types. The magnitude of the variation of hydrological response is tied to the magnitude of the changes in land cover. De Roo et al., (2001), for instance, investigated the effects of land use changes on floods in two European catchments and identified different results depending on the magnitude of the land cover change. It is however true that the effect of land use has in general a low influence on river discharge; even if we expect this influence to be locally high in areas which have experienced a heavy urbanisation. We aim to investigate this more in detail in the future.

According to LISFLOOD design, large changes in water abstraction can have significant impacts on surface water resources. LISFLOOD makes use of the concept of water region, defined as the area including the locations of water demand and abstraction. Water for domestic use, flooded irrigation (rice) and other irrigation types, industrial use, cooling, and livestock can be abstracted from rivers, lakes, reservoirs (provided that environmental flow and volumes are preserved). A considerable increase in water abstraction in a water region can diminish surface water resources within the same area. It is here noted that, in case of water scarcity, LISFLOOD prioritizes domestic use over the other usages. LISFLOOD also accounts for groundwater abstraction for human use, except for flooded irrigation and cooling processes. Increased groundwater abstraction can locally reduce (or halt) baseflow.

We will add a summary of the answer to the section dedicated to dynamic land use and water abstraction.

P25-L592-94: Here, maybe include a sentence that for long-term trends this needs to be considered and has to be included in any interpretation of results.

R: We will rewrite significant parts of the discussion in lights of the extra validation steps that we will perform (See general reply).

P25-L601: car -> can

R: Change will be implemented.

P26-L614f: Do you think that one month spin-up is enough?

R: After verifications, the lower zone is initialized using the 71 years PreRun. The reason causing the initial months to be unreliable is the fact that water volumes at time 0 in the channels are not known and the model sets a conventional initial volume (LISFLOOD uses half-bankful). This state is expected to "quickly" adjust (it is much shorter memory compared to soil and groundwater). The amount of time required to remove the impact of the initial, fictitious value depends on mainly on river geometry and climate. After additional verifications, we decided to remove the full year of 1950.

Reference list:

Cannon, A. J.: Multivariate quantile mapping bias correction: an N-dimensional probability density function transform for climate model simulations of multiple variables, Clim. Dynam., 50, 31–49, https://doi.org/10.1007/s00382-017-3580-6, 2017.

Cantoni, E., Tramblay, Y., Grimaldi, S., Salamon, P., Dakhlaoui, H., Dezetter, A., and Thiemig, V.: Hydrological performance of the ERA5 reanalysis for flood modeling in Tunisia with the LISFLOOD and GR4J models, Journal of Hydrology: Regional Studies, 42, 101169, https://doi.org/10.1016/j.ejrh.2022.101169, 2022.

De Roo, A., Odijk, M., Schmuck, G., Koster, E., and Lucieer, A.: Assessing the effects of land use changes on floods in the meuse and oder catchment, Physics and Chemistry of the Earth, Part B: Hydrology, Oceans and Atmosphere, 26, 593–599, https://doi.org/10.1016/S1464-1909(01)00054-5, 2001.

Fortin, F.-A., De Rainville, F.-M., Gardner, M.-A., Parizeau, M., and Gagné, C.: DEAP: Evolutionary Algorithms Made Easy, Journal of Machine Learning Research, 13, 2171–2175, 2012.

Gupta, H. V., Kling, H., Yilmaz, K. K., and Martinez, G. F.: Decomposition of the mean squared error and NSE performance criteria: Implications for improving hydrological modelling, Journal of Hydrology, 377, 80–91, https://doi.org/10.1016/j.jhydrol.2009.08.003, 2009.

Lange, S.: Trend-preserving bias adjustment and statistical downscaling with ISIMIP3BASD (v1.0), Geoscientific Model Development Discussions, 1–24, https://doi.org/10.5194/gmd-2019-36, 2019.

**REVIEWER 2**

Dear Editor,

In the manuscript *"HERA: a high-resolution pan-European hydrological reanalysis (1950-2020)"* the authors present a high spatial (1 arc-degree) and temporal (6 hourly) European discharge dataset based on simulations with the LISFLOOD model. Furthermore, the authors validate the dataset by comparing simulated discharge with discharge observations throughout Europe and comparing performance over various periods, catchment areas, reservoir impact ratios, and flow percentiles. The manuscript is clear and concise and results are validated in detail. Moreover, all datasets are well-distributed and publicly available and accessible. Nevertheless, some additional details are needed to better evaluation of the simulation results and additional recognition of other works is needed to put this study in a broader context. Therefore, I would recommend **major revisions** (close to minor revisions) for this manuscript. Below is a more expansive description of my main arguments as well as a list of line-by-line comments.

**General answer:**

R: We want to thank Anonymous Referee #2 for his encouraging review and for providing a list of related and relevant references, which we will include in out revised manuscript.

**Additional details**

Although the manuscript methods and results are generally clear, some additional details are needed to sufficiently evaluate the dataset performance.

C: First, the authors mention they calibrated the model (lines 156-176). However, some more detail on the calibration is needed. Did the authors calibrate the model on the same stations on which the validation is performed? If so, can the authors provide any indication on how the model performs in ungauged basins? This information is important as the main aim of the manuscript dataset is to provide discharge in regions/periods where observations are not available.

R: As stated in Section 3.1: the set of stations for the validation was compiled independently from the one used in the EFAS calibration. However, the two datasets have 1026 stations in common. This will be made clearer in Section 2.1.2 (calibration). Having different stations does not necessarily mean that the catchments which were not used in calibration do not have calibrated parameters. The extent of calibrated catchments is displayed in Figure S1. Most of the stations used in the validation (Figure S2) fall within the calibrated area. We agree with the reviewer that performances in ungauged catchments is very relevant. We will prepare an extra plot similar to the ones displayed in Figure 7 showing performance in: 1) stations used in calibration, 2) stations within the calibrated domain 3) stations outside the calibrated domain (regionalized parameters).

C: Second, the authors mention that the model includes human water abstractions (lines 307-331). However, results for the actual human water withdrawals are never mentioned. Although these abstractions are not the main focus of the manuscript, they can give a good insight into the intermediate processes in between precipitation and discharge. A table with the aggregated human water abstractions (and possible other water-balance components such as evapotranspiration) in the supplementary would be sufficient.

R: It is not clear what the reviewer is asking. A table summarizing water abstraction over time? Figure 4.c is already providing this information. We can transfer this into a table.

C: Third, the authors exclude some suspicious (i.e. poorly performing) observation stations from the final validation because they may result from incorrect pre-processing (lines 364-374). However, it is

not shown if this is actually the case. Therefore, it might be that good simulation performance can only be achieved because poorly performing stations are removed. Although the limited number of removed stations indicates this is likely not the case, could the authors also provide some of the validation results with these stations included (e.g. in the supplementary)?

R: Each station which was excluded through this "poor performance" filtering was manually checked by the first author. We will add to the supplement the table with removed station and the reason for removal. The main reason for the station removal are the provided in the manuscript and are the following: wrong spatial match, erroneous station location, and doubtful observations. Most of the time, stations are removed because they are located on a stream with an upstream area<100km$^2$ near its confluence with a larger river.

**Additional recognition**

C: Although the manuscript sufficiently indicates the need for high-resolution discharge simulations, there is insufficient recognition of (and comparison with) other efforts in the hydrological modeling domain. For example, the authors mention that other models usually run at a coarse spatial resolution (lines 53-55). However, over the last decades substantial efforts have been made to increase spatial resolution (Wood et al., 2011; Bierkens et al., 2015; Wada et al., 2017; Hanasaki et al., 2022), and other models have made regional to global simulations that even exceed the spatial resolution presented here (e.g. O'neill et al., 2021; Aerts et al., 2022; Hoch et al., 2023; van Jaarsveld et al., 2024).

In addition, comparisons with other studies should be made. The authors mention that such comparisons are difficult (lines 565-568) as simulation inputs varies widely. However, without such comparisons one cannot assess the dataset performance in the context of the larger hydrological field and the presented dataset is little more than 'another model simulation'.

- Wood, E. F., Roundy, J. K., Troy, T. J., Van Beek, L. P. H., Bierkens, M. F., Blyth, E., ... & Whitehead, P. (2011). Hyperresolution global land surface modeling: Meeting a grand challenge for monitoring Earth's terrestrial water. *Water Resources Research*, *47*(5).

- Bierkens, M. F., Bell, V. A., Burek, P., Chaney, N., Condon, L. E., David, C. H., ... & Wood, E. F. (2015). Hyper-resolution global hydrological modelling: what is next? "Everywhere and locally relevant". *Hydrological processes*, *29*(2), 310-320.

- Wada, Y., Bierkens, M. F., De Roo, A., Dirmeyer, P. A., Famiglietti, J. S., Hanasaki, N., ... & Wheater, H. (2017). Human–water interface in hydrological modelling: current status and future directions. *Hydrology and Earth System Sciences*, *21*(8), 4169-4193.

- Hanasaki, N., Matsuda, H., Fujiwara, M., Hirabayashi, Y., Seto, S., Kanae, S., & Oki, T. (2022). Toward hyper-resolution global hydrological models including human activities: application to Kyushu island, Japan. *Hydrology and Earth System Sciences*, *26*(8), 1953-1975.

- O'neill, M. M., Tijerina, D. T., Condon, L. E., & Maxwell, R. M. (2021). Assessment of the ParFlow–CLM CONUS 1.0 integrated hydrologic model: evaluation of hyper-resolution water balance components across the contiguous United States. *Geoscientific Model Development*, *14*(12), 7223-7254.

- Aerts, J. P., Hut, R., van de Giesen, N., Drost, N., van Verseveld, W. J., Weerts, A. H., & Hazenberg, P. (2022). Large-sample assessment of varying spatial resolution on the streamflow estimates of the wflowsbm hydrological model. *Hydrology and Earth System Sciences*, *26*(16), 4407-4430.

- Hoch, J. M., Sutanudjaja, E. H., Wanders, N., Van Beek, R. L., & Bierkens, M. F. (2023). Hyper-resolution PCR-GLOBWB: opportunities and challenges from refining model spatial resolution to 1 km over the European continent. *Hydrology and Earth System Sciences*, *27*(6), 1383-1401.

- van Jaarsveld, B., Wanders, N., Sutanudjaja, E. H., Hoch, J., Droppers, B., Janzing, J., ... & Bierkens, M. F. (2024). A first attempt to model global hydrology at hyper-resolution. *EGUsphere*, *2024*, 1-32.

R: Thank you very much for this comment. We were not aware of most of the studies you mention in your comment. We are grateful you suggested these additional articles and will read these carefully. References will be included into the introduction and discussion when relevant.

Regarding the comparison with other studies. We agree that is a critical aspect for both the authors and the community to compare different models and modelling avenues. During the writing on the article, we did not find any comparable dataset that was publicly available. Among the studies you suggested, one seems to be ideal for the comparison: Hoch et al., 2023. Unfortunately they did not make the simulated data available. We contacted the first author of this study, without response at the moment. Nevertheless, we identified a comparable model run with the mHM model (https://mhm-ufz.org/). More in particular, the model run was used in the following publication: Tarasova et al. (2023). All the information on model forcing, calibration and validation can be found in the Method section of the article. It is important to note here that this other simulation is comparable, but different in many regards (resolution: 1.8km vs 5km, forcings: E-OBS vs Era5-land bias corrected, model: OS LISFLOOD vs mHM). The authors kindly agreed to send us a sample of the simulated discharge over 1340 locations across Europe. We will then add a comparison with this dataset over the period 1960-2010.

**Line-by-line comments**

C: Some additional focus on the abstract, as this could be improved

R: we will work on improving the abstract

C: Lines 11-12 "European rivers have been put under pressure (…) resulting in changes in climate (…)": Rephrase, pressure on European rivers does not result in changes in climate, etc.

R: change will be implemented.

C: Lines 12-13 "environmental conditions": What are environmental conditions?

R: The ones mentioned just before: "climate, land cover, soil properties and channel morphologies". We will clarify this in the revised manuscript.

C: Line 15 "attribute": Attribute to what?

R: We meant attribute changes to drivers such as climate, land use, water abstraction changes.

C: Line 20 "with weather observations": If you mean EMO-1k here, EMO-1k is not just weather observations (includes interpolation).

R: We will change the sentence to stress that EMO-1 an observation-based gridded dataset.

C: Line 21 "surface fields": Unclear what surface fields mean. Just maps?

R: Yes, it is how their creators call these maps, which represent, as stated in the same sentence, catchment morphology, vegetation, soil properties, land use, water demand, lakes…

C: Line 15: Please include some evaluation metrics here.

R: I guess you meant line 25. Yes I we will add median/mean bias and/or KGE'.

C: Line 29: As mentioned above, other studies have also done high-resolution hydrological simulations for Europe.

R: Yes, we will acknowledge these other studies in the revised manuscript.

C: Line 37 "IPCC": Please cite the IPCC chapter (and its authors).

R: change will be implemented.

C: Line 40 "to understand": More philosophical note, but simulating hydrology does not give us more *understanding*. Rather, understanding goes into the model processes and the model simulation gives us an idea of the emerging properties. Maybe do not use the word understanding.

R: I agree with you, however the sentence is not specifically about hydrological modelling. I can replace the word understand to avoid confusion.

C: Line 67: ERA-5 lands needs citation.

R: change will be implemented.

C: Line 71 "they are not observations": why this sentence?

R: This sentence is here as a disclaimer, stressing the fact that climate reanalysis are not observations and therefore cannot be considered as "ground truth".

C: Lines 76-89: The section feels disconnected from the rest of the chapter.

R: The main objective of this paragraph is to introduce OS LISFLOOD and the recent developments towards "hyper-resolution" hydrological modelling. I think it is relevant in the introduction. I am unsure what changes can be done here.

C: Lines 115-135: Human impacts such as withdrawal/consumption are not mentioned here.

R: I am sorry but I don't understand what that comment implies for the manuscript.

C: Lines 151-152: What happens to the pixels with smaller upstream areas? Are these not simulated or not included in the output?

R: OS LISFLOOD is a spatially distributed model. Each cell (1 arcmin x 1 arcmin) contains discharge information in the output. Including all the cells in our final product would result in very heavy files containing a lot of cells with zero discharge. We know that in GHM like LISFLOOD, performances tend to decrease for smaller catchments (also the case for HERA as shown in Figure 7.b). We then decided to keep cells based on a threshold on upstream area, in that case 100 km$^2$, which roughly represents 30 grid cells.

C: Lines 156-176: What calibration method did you use?

R: In order to calibrate LISFLOOD, 1903 sub-catchments were used, each of which is closed by a station that has at least four equivalent years' worth of discharge observations within the period from 1990 to 2021. The parameter values of these catchments were identified using the Distributed Evolutionary Algorithm for Python (DEAP, Fortin et al. 2012). DEAP was used to explore the

parameter space and identify the parameter set leading to the highest value of the modified Kling Gupta Efficiency (KGE', Gupta et al., 2009).

The calibration protocol went from head-catchments to downstream catchments in a top-down manner, prescribing physical dependencies between upstream and downstream catchments within the same basin. Depending on the length of available discharge observations, each station's calibration period varied, but it was never less than four equivalent years. If a time period longer than eight years was available, the observed discharge record was divided in half for calibration and verification purposes. The calibration was performed using the most recent period, as it more accurately represents the hydrological and climatic conditions that are used in EFAS operational forecasts.

This combined calibration approach delivered 14 parameter maps with pan-European extent whose purpose is to best represent natural processes such as snow melt, water infiltration into the soil, surface water flow, groundwater flow, lakes, and reservoir dynamics. We will complement Section 2.1.2 with these information.

C: Lines 191-214: Would be very nice if you have some comparisons between your corrected ERA5-land simulation and the EMO-1k simulation. What are the differences? To what extent does ERA5-land add value (except for the longer period)?

R: It is not clear to us what you are asking for in this comment, comparison maps of meteo datasets? There is obviously no perfect grid cell-by-grid cell match, only the distributions over time are similar. We want to stress that the only reason we use ERA5-Land is because it is available for a longer period and used for the entire run for consistency of the approach.

C: Line 248 and 250 "CEMS_SurfaceFields_2022": Please find a better name in the text.

R: Unfortunately, We did not choose the name of the dataset, its authors defined it, the article preprint is available here: https://egusphere.copernicus.org/preprints/2023/egusphere-2023-1306/

C: Figure 4: Please provide these numbers in volumes (km3 year-1) to allow the reader to compare. Also please include the irrigation water demands.

R: Irrigation water demand is computed within the model based on the difference between potential transpiration (Tmax) and actual transpiration (Ta). The computation of Tmax and Ta is specific to every crop and is described in details in the dedicated OS LISFLOOD github page: https://ec-jrc.github.io/lisflood-model/2_07_stdLISFLOOD_plant-water-uptake/

Furthermore, irrigation water demand for rice is computed separately: https://ec-jrc.github.io/lisflood-model/2_17_stdLISFLOOD_irrigation/

It is not straightforward to retrieve these variables from OS LISFLOOD, and we did not do it during the generation of HERA. We are therefore unable to provide irrigation water demand along with other water demand. We can however redo the plot with numbers in km3.y-1.

C: Lines 421-422 "and their deviation from the (…)": What does this sentence mean? Does it just say that the performance is poor because it deviates from regions (and climates) where the performance is good?

R: Yes, that is more or less what the sentence means, although you can appreciate the tone of our sentence "Factors that can explain the poor performance […] include" that differs from the one of your interpretation. The point here was to bring knowledge about general performances of OS LISFLOOD, which is usually better in a given hydro-climatic window, here corresponding to temperate climates. We will modify the sentence to improve its clarity.

C: Line 424 "1.2es": ?

R: It is bias, no idea how it became 1.2es…We will correct this.

C: Lines 444-446: Would be nice, at some point (maybe not for this paper), to run another simulation without the varying surface fields and only with a change in climate and repeat this validation.

R: Very good idea. It is indeed in the pipes, and we hope to be able to perform the validation as well on that run.

C: Lines 480-497: Although the poor performance might be attributed to errors and biases in the inputs or observations, they may also bet attributed to the model's processes description and implementation.

R: Yes, it is true. We could not quantify here the contribution of different sources of uncertainty that we mention in the manuscript: *"These deviations can be attributed to errors or biases in our climate inputs (McClean et al., 2023), in the hydrological model (Feyen and Dankers, 2009), but also to errors in flow measurements, especially for Q05 (Despax, 2016; Tomkins, 2014) and anthropogenic impacts on low and median flow regimes (Brunner, 2021) that are not accurately represented in the model".* There is certainly a contribution of the model's processes description and implementation to the final uncertainty.

C: Lines 599-600 "Uncertainties inherent to model structure have long been overshadowed": Also in this manuscript, what is the purpose of this sentence?

R: The purpose of this sentence was mainly to highlight that uncertainties inherent to model structure have been rarely assessed in the past, leaving us short of options to assess the uncertainties inherent to OS LISFLOOD is that run. We will make the sentence clearer.

C: Line 659 "realistic": you have not shown that this inclusion makes it more realistic, better to use "detailed".

R: We agree and will implement this change.

**References**

Fortin, F.-A., De Rainville, F.-M., Gardner, M.-A., Parizeau, M., and Gagné, C.: DEAP: Evolutionary Algorithms Made Easy, Journal of Machine Learning Research, 13, 2171–2175, 2012.

Gupta, H. V., Kling, H., Yilmaz, K. K., and Martinez, G. F.: Decomposition of the mean squared error and NSE performance criteria: Implications for improving hydrological modelling, Journal of Hydrology, 377, 80–91, https://doi.org/10.1016/j.jhydrol.2009.08.003, 2009.

Tarasova, L., Lun, D., Merz, R., Blöschl, G., Basso, S., Bertola, M., Miniussi, A., Rakovec, O., Samaniego, L., Thober, S., and Kumar, R.: Shifts in flood generation processes exacerbate regional flood anomalies in Europe, Commun Earth Environ, 4, 49, https://doi.org/10.1038/s43247-023-00714-8, 2023.

**REVIEWER 3**

Dear Editor,

In the enclosed manuscript, the authors present a high spatial and temporal resolution river discharge reanalysis dataset which covers the European region for the years 1950 to 2020. The objectives are clearly stated and the results are well presented. In addition, the authors show the outcomes of an extensive validation and evaluation of the data they put forward. The dataset is easily accessible and the accompanying usage notes are clear.

**Specific comments**

There are five points that need addressing that will improve the overall quality of this manuscript. The first topic relates to the general framing of this work in relation to previous work done on high resolution hydrological models. The authors indicate that this is a first-of-its kind dataset (line 29,100, 651-652), however previous work by Hoch et al 2023 provides results at resolutions finer than that presented here for this modelling domain. The authors should acknowledge this work in the introduction, discussion and conclusion.

First, we want to thank Anonymous Referee #3 for this review and the valuable feedback. We were not aware of the work you are mentioning while writing the manuscript. We are grateful that you and Anonymous Referee #2 suggested this article and others. We will read this article carefully. References will be included into the introduction and discussion when relevant.

Unfortunately, Hoch et al. (2023) did not make their simulated data available. It would be an interesting exercise to compare the output of their highest resolution run with HERA. We contacted the first author of this study, without response at the moment.

**References:**

Hoch, J. M., Sutanudjaja, E. H., Wanders, N., van Beek, R. L. P. H., and Bierkens, M. F. P.: Hyper-resolution PCR-GLOBWB: opportunities and challenges from refining model spatial resolution to 1 km over the European continent, Hydrology and Earth System Sciences, 27, 1383–1401, https://doi.org/10.5194/hess-27-1383-2023, 2023.

The second point relates to the authors decision to retain only cells where the upstream area exceeds 100km$^2$. The title, abstract and conclusion refer to a "pan European" product. Yet, given that many cells are excluded this claim becomes questionable.

R: As you can see in Figure 2, the dataset indeed covers the whole Europe, as we claim. Including all the cells in our final product would result in very heavy files containing many cells with zero discharge. Furthermore, we consider that 100km2 is already fairly detailed for a continental-scale dataset. We detail a bit more the rationale behind the exclusion of grid cells with small upstream area in following replies.

Fig 2: The inclusion of rivers with an upstream area > 100km$^2$ outside of the domain for which data is made available is initially misleading. For clarity, perhaps it is more clear if these are omitted.

R: There were debates between co-authors about what to represent on this figure. We decided to show all grid cells the discharge data that was generated, even the ones outside of the domain. Many of these pixels were needed to correctly simulate river discharge in Baltic countries for instance.

Line 151-152: What is the rationale for retaining only those pixels where the upstream area is greater than 100km$^2$? This should be detailed in the manuscript. In addition, what percentage of the domain is excluded because of this, it is difficult to see from the information presented in Figure 2.

R: The reasons for exclusions of pixels with small upstream area are the following: Including all the cells in our final product would result in very heavy files containing a lot of cells with zero discharge. We also know that in GHM like OS LISFLOOD, performances tend to decrease when going to smaller catchment (also the case for HERA as shown in Figure 7.b). We then decided to keep cells based on a threshold on upstream area, in that case 100 km$^2$, which roughly represents 30 grid cells. By doing this,

we do not "exclude" information from the dataset; as discharge in grid cells with larger upstream area is a result of the accumulation of upstream discharge from smaller pixels. All pixels shown in Figure 2 are in the dataset.

The third point relates to the interesting result that model performance increases over time. The authors provide viable explanations for this result on lines 590-599. Yet, given that the authors state on line 614 -616 that the discharge values for the start of 1950 could be inaccurate; an alternative explanation for this result could be related to unstable initial conditions? If the values produced from the 71 year pre-run are not at equilibrium, an increasing model performance over time can also be expected as the model reaches true equilibrium states. The discussion would benefit from including this line of argument.

R: Thank you for this comment. After discussions among co-authors, we can discard the convergence towards equilibrium as a factor explaining converging performances. From past and ongoing experiments, we notice that the convergence can mainly have an impact on the third soil layer of the model while we did not see a significant impact on discharge. Nonetheless, there is a reason to remove the first months of the simulation, which is the related to how rivers are initialized in OS LISLFOOD. In the absence of information at time 0, the model initiates rivers at a conventional value corresponding to half-bankful volume, which can lead to rivers being full/too empty. To prevent this initialization to have an influence on the reanalysis, we decided to remove the whole 1950 year from the dataset, which now starts on January 1st 1951. We will add a sentence in the discussion mentioning that point.

Line 614 -616: From what dates do the discharge values become accurate? If the authors suggest that some of the published data should not be used would it not be more clear if the inaccurate data where not published at all and instead present the community with data that can be used directly?

R: This is directly related to our previous answer. The effect of the river initialization is very heterogeneous amongst catchments as fictitious value depends on mainly on river geometry and climate. We agree with you regarding the fact that is clearer to remove inaccurate data. From experience and after additional verifications, we concluded that removing the first year of simulation was a safe and simple way to remove potential inaccurate data.

The fourth point that needs addressing is that the manuscript would be strengthened if more information on how the data described in lines 307 – 321 are used for aggregation and disaggregation of the water withdrawal inputs (line 314).

R: The derivation of the gridded water withdrawal data involved numerous processing steps and input datasets. We agree that a clear and complete description of the data and methods is important; however, due to limited space, we are unable to explain the full procedure. Therefore, we refer to Huang et al. (2018) and Choulga et al. (2023a) in the paper. For even more details, please refer to the public GitHub repository (https://github.com/hylken/lisflood-water-demand-historic), which includes a detailed README file and the actual code used to generate the water withdrawal data. However, after re-reading the paragraph in question, we removed the phrase "a variety of datasets. These include" to avoid ambiguity. We have copied the revised paragraph below.

*"Human water use, representing water withdrawal from the natural environment (e.g., rivers, reservoirs, groundwater) for human needs, is grouped in four main sectors: livestock, domestic, manufacturing industry, and energy production. Within OS LISFLOOD, water is supplied by surface water bodies and groundwater depending on the sector (Choulga et al., 2023a). To derive monthly historic sectoral water withdrawal maps, we followed the methodology of Huang et al. (2018) and used the Food and Agriculture Organization (FAO) AQUASTAT sectoral water withdrawal data (FAO, 2023) as a starting point. These data were subsequently spatially and temporally disaggregated using he Global Human Settlement Layer (Schiavina et al., 2019; Florczyk et al., 2019) for population estimates, the Global Change Analysis Model (GCAM; Calvin et al., 2019) for regional water withdrawal and electricity consumption, and the Gridded Livestock of the World (GLW; Gilbert et al., 2018) for livestock distribution. Additional datasets included the Multi-Source Weather (MSWX; Beck et al., 2022) for air temperature data, United States Geological Survey (USGS) water withdrawal estimates, and Vassolo and Döll (2005) industrial and thermoelectric withdrawal maps. More information on water demand and input datasets used is provided in Choulga et al. (2023a)."*

Last, the authors used computed KGE scores as a criteria for excluding GRDC stations (line 368 – 373). Excluding stations where the distance is greater than 2.5km and have poor scores implies that stations that have a distance discrepancy greater than 2.5km and have good scores are retained. This could result in an inflation of model performance due to the erroneous inclusion of false positive results. The authors should consider using the distance discrepancy regardless of KGE score as an exclusion criteria.

R: This is true that 25 stations are removed with this procedure. First, we want to remind here that the initial step of the spatial matching implies to only retain the 9 pixels around the location of the river gauge. We can therefore roughly estimate what would be the maximal distance between the considered station and the furthest pixel centre. With few steps of simple geometry we find that, at worst, the distance between the station and a matched pixel could be ≈3.8 km (1.8*sqrt(2) *1.5). This means that distance is already used as an exclusion criterion. Secondly, I do not understand how keeping points in a 4km radius (knowing grid resolution is 1.8 x 1.8 km) that show acceptable skill would be problematic. Most likely, this would lead to the river pixel being upstream or downstream the station. The criteria related to low KGE' performance was added to remove situations where the wrong river is matched with the station. I do not expect the KGE' to be higher on a river which is not the one of the station.

**Technical corrections**

HER needs to be replaced with HERA in figure legends for Fig 4, Fig 6 and Fig 7.

R: It is HERA, we will remove all instance of HER.

Full stops are missing in lines 291, 367, and 489.

R: change will be implemented.

Line 291: "used ,we". Incorrect space between used and the comma.

R: change will be implemented.

Line 597: "..219).It" insert space.

R: change will be implemented.

Some of the intext references are not present in the bibliography and should be updated.

R: We will carefully work on the reference list for the revised manuscript.

**Additional comments to reviewers and editor:**

We want to mention here a couple of changes that were not mentioned in the initial replies to reviewers:

- **Addition of a new co-author:**
  HERA relies on the calibration of OS LISFLOOD done in the context of EFAS v5.0. Reviewers asked for more details about the calibration, which we added in the revised manuscript. This was done thanks to the help of Cinzia Mazzetti, who was in charge of this calibration at ECMWF. She was therefore added to the co-authors for her contribution.

- **Modification of the station matching methodology:**
  The methodology to match HERA river cells with hydrometric stations was revised, to make it clearer. Only stations with a long enough record were kept (more than 30 years). This new method was not mentioned in the reply to reviewers.

- **Extended supplementary material:**
  The supplementary material was enriched with the improvement of Figure S1, the addition of Table S4 and Table S5. A new section comparing HERA and another European hydrological simulation (EUmHM) (Figure S3, Figure S4, Figure S5, Figure S6).

- **Corrected Figure 9:**
  The violin plot of Figure 9 has been modified as the previous version was erroneous.

---

## Author Response (AR3)

Reviewer: The manuscript revision is an improvement from the initial manuscript. The revision remains clear and the dataset is well-distributed (i.e. publicly available and accessible). However, the lack of recognition and comparison against other works, as mentioned in the previous reviews (reviewers #2 and #3) remains. Also, the writing in the manuscript should still be improved. Therefore, I would again recommend major revisions for this manuscript. Below is a more expansive description of my main arguments as well as a list of line-by-line comments.

Additional comparison and recognition

Based on the previous round of reviews, the authors have mentioned other similar works in the introduction/discussion and included a thorough harmonized comparison with results from the mHM model.

Nevertheless, the context of the efforts toward hyperresolution modeling remains very limited. Especially in light of the previous round of reviews, the authors use overly strong language when presenting their dataset, aiming to indicate this dataset is a first of its kind. However, several other studies have done similar work. Although this dataset contains novel aspects, there are also downsides (i.e. limited spatial coverage and resolution). Therefore, this dataset should be placed within the context of and compared to other work in the area of hyperresolution modeling.

Note that comparison does not only entail a thorough harmonized comparison as done with the mHM model (especially as the authors could not acquire the outputs from some other studies). Rather, some context on how other studies handle their hydrological reanalyses (i.e. spatial coverage, spatial resolution, temporal coverage, socioeconomic conditions; see Line 647-650), and whether their reported performance is in line with HERA. Note that even though other studies (e.g. GLOFAS-ERA5) use different inputs, their outputs can still be compared.

Reply from authors: We appreciate the reviewer's feedback on the revised manuscript, most of which will improve its quality. We have indeed included references to other high-resolution hydrological reanalysis studies and compared our dataset, HERA, with a run of the mHM model. However, we would like to clarify the statement about HERA being a first-of-its-kind dataset. It refers to the fact that we could not find a publicly available hydrological dataset with characteristics that make a comparison meaningful. This means with comparable (i) resolutions (GLOFAS-ERA5 is 0.1°, meaning that 1 pixel is 30 times larger than HERA), (ii) spatial coverage (Europe), (iii) temporal coverage ( HERA starts is 1951, 28 years earlier than GLOFAS-ERA5). We would appreciate any information about publicly available datasets that have characteristics comparable to HERA.

Regarding GLOFAS-ERA5 more in particular, we acknowledge it in the manuscript, and it indeed has a lot in common with HERA (ERA5 as a primary meteorological forcing, LISFLOOD as hydrological model, KGE' used as a skill metric in calibration). We however believe that the two reanalysis, and their validation work, are too different to be compared. GLOFAS-ERA5 is a global reanalysis, mainly developed for large rivers. Most catchments used in its validation have an area above $10\,000$ km$^2$ while the large majority of catchments used in the validation of HERA have an upstream area below $10\,000$ km$^2$. Performances of GLOFAS-ERA5 are very well described in Harringan et al. (2020), and that article was actually an inspiration for working on a much more detailed reanalysis product with focus on Europe. Although the present article does not exactly reproduces the same figures, same metrics are

used, making a quick comparison simple. As stated before, we are doubtful that this comparison makes sense since the catchment sets used to validate both datasets are too different. We believe that the mHM run we used for comparison in the supplement is more relevant in terms of (i) resolutions (ii) coverage, (iii) length and (iv) purpose of the dataset.

We addressed most of the specific comments raised by the reviewer, resulting in the following main changes:

- Addition of a section in the supplement with a table providing a summary of reported performances of HERA, GLOFAS-ERA5, GRFR and EUmHM.
- Modification of the discussion to incorporate reviewer's suggestions
- Addition of new literature.
- Correction of typos and clarification of vocabulary (all manuscript)

Line number in authors' reply refer to the update Track Changes document.

Specific comments (note that line numbers refer to the markup document)

Line 15 '(…) anthropogenic activities have altered (…), soil properties, channel morphologies (…)': these are not changed in this study, which may confuse.

R: We understand the reviewer's concern that the phrase "anthropogenic activities have altered (…) soil properties, channel morphologies" may be misleading since these aspects are not specifically addressed in this study. However, we believe that the abstract's introductory sentence aims to provide a general context for the research, rather than a detailed description of the study's focus. The phrase serves to highlight the importance of hydrological reanalyses in understanding the impacts of human activities on the environment, which is relevant to the broader field of study.

Line 17-19 'The availability of consistent estimates of river flow at global and continental level is a necessity to assess and attribute changes in the hydrological cycle.': one cannot attribute changes based on consistent estimates of river flow, rather several simulations with different forcings/parameters are needed to attribute changes. Although this is possible with the model setup in this study, this study presents a dataset. See also Line 38-40.

R: We removed "and attribute" from the sentence.

Line 37-38 'HERA is the first publicly available long-term, high-resolution hydrological reanalysis for Europe': since there are other highER resolution hydrological reanalysis available for Europe, remove sentence.

R: We appreciate the reviewer's suggestion to remove the sentence stating that HERA is the first publicly available long-term, high-resolution hydrological reanalysis for Europe. However, our statement is based on our thorough search for publicly available datasets with similar characteristics to HERA. Despite our efforts, we were unable to find other publicly available long-term, high-resolution hydrological reanalyses for Europe.

We are aware of the EFAS v5 long run, which shares similarities with HERA, but its shorter temporal coverage (30 years) led us to consider it as a different type of dataset. We would be happy to revise the sentence if the reviewer can provide us with examples of publicly available datasets that meet the same criteria as HERA (long-term, high-resolution, and publicly available).

Line 38-40 'Despite its limitations, it enables (…)': double 'it', rephrase to 'Despite its limitations, HERA enables (…).

R: Sentence changed.

Line 45-47 'These evolving conditions have significantly changed flows in European streams and rivers (…)': requires citation.

R: The following references have been added:

Barker, L. J., Hannaford, J., Parry, S., Smith, K. A., Tanguy, M., and Prudhomme, C.: Historic hydrological droughts 1891–2015: systematic characterisation for a diverse set of catchments across the UK, Hydrology and Earth System Sciences, 23, 4583–4602, https://doi.org/10.5194/hess-23-4583-2019, 2019.

Gudmundsson, L., Boulange, J., Do, H. X., Gosling, S. N., Grillakis, M. G., Koutroulis, A. G., Leonard, M., Liu, J., Müller Schmied, H., Papadimitriou, L., Pokhrel, Y., Seneviratne, S. I., Satoh, Y., Thiery, W., Westra, S., Zhang, X., and Zhao, F.: Globally observed trends in mean and extreme river flow attributed to climate change, Science, 371, 1159–1162, https://doi.org/10.1126/science.aba3996, 2021.

Vicente-Serrano, S. M., Peña-Gallardo, M., Hannaford, J., Murphy, C., Lorenzo-Lacruz, J., Dominguez-Castro, F., López-Moreno, J. I., Beguería, S., Noguera, I., Harrigan, S., and Vidal, J.-P.: Climate, Irrigation, and Land Cover Change Explain Streamflow Trends in Countries Bordering the Northeast Atlantic, Geophysical Research Letters, 46, 10821–10833, https://doi.org/10.1029/2019GL084084, 2019.

Wang, H., Liu, J., Klaar, M., Chen, A., Gudmundsson, L., and Holden, J.: Anthropogenic climate change has influenced global river flow seasonality, Science, 383, 1009–1014, https://doi.org/10.1126/science.adi9501, 2024.

Line 45-47 'These evolving conditions (…), leading to challenges for hydrological sciences, related, for example, to long term variability, climate change, extremes or human alterations of the water cycle': How do changes in the water cycle challenge hydrological sciences?

R: For example:

- Flood protections design: if flood magnitudes are on the rise, societies should reassess the design of flood protection infrastructures and dams spillways. This pushes hydrological scientists into adapting there methods, by using non-stationary statistics for example.
- Water availability: if rivers tend to dry, as it is the case in southern Europe, less water will be available for human usages, and could therefore create further conflicts around water use (socio-hydrology).
- Understanding the complex response of diverse catchments to climate change is also a challenge for hydrological sciences.

Line 49-51 'Observations (…) are lacking at a high enough spatial density (…)': lacking for what?

R: We have update this sentence, now indicating that this hampers analysing pan-European long term trends: "*Observations, despite continuous improvements (Blöschl et al., 2019a; Ekolu et al., 2022), can hamper the analysis of Pan-European long-term trends due to sparse spatial distribution in some regions and temporal discontinuities*".

Line 66-68 'Remote sensing technologies now provide high resolution input for hydrological models (…)': Inputs at the spatial resolution of this study has been provided already for quite some time.

R: We agree with the reviewer. Technologies are rapidly evolving and provide increasingly more accurate and higher resolution information. We updated the sentence accordingly.

Line 76-78 '(…) numerous homogeneous environmental variables (…)': what does this mean?

R: The formulation was not optimal. We have now rephrased this: "Reanalysis products typically provide a large number of variables (e.g., precipitation, wind speed, temperature) that are physically consistent with homogeneous spatiotemporal resolution."

Line 89 'Section 2.2': for me, all sections are unnumbered, so these references (throughout the document) do not help.

R: Yes, we apologize for that. The uploaded revised manuscript has the sections correctly numbered.

Line 104 'Therefore, tis': 'Therefore, this'

R: This has been corrected.

Line 110: 'improvements in processing speed, spatial and temporal resolutions, calibration': 'improvements in processing speed, spatial and temporal resolutions and calibration'

R: This has been corrected.

Line 115-113 'These developments make this dataset the first publicly available long-term Pan-European hydrological reanalysis taking into account the evolving socioeconomic conditions that have altered the hydrological cycle since 1951': since there are other hydrological reanalysis available that take into account the evolving socioeconomic conditions (with explicit results for, for example, human water withdrawals) for Europe, remove sentence.

R: We would appreciate clarification and supporting evidence regarding the existence of publicly available Pan-European hydrological reanalyses that account for evolving socioeconomic conditions, as this would inform our assessment of the novelty and uniqueness of the dataset presented in this work.

Line 170-171 '(…) a 71-year pre-run (longest possible period)': would the spinup not be able to loop the 71 years to get a longer period?

R: The reviewer has a point, in principle, yes. We therefore removed "longest possible period". We believe a 71-year pre-run is sufficiently long.

Line 178 'River initialization in OS LISFLOOD can lead to unrealistic discharge in some catchments': why is this the case after a 71-year spinup?

R: The reason causing the initial months to be unreliable is the fact that water volumes at time 0 in the channels are not known and the model sets a conventional initial volume (LISFLOOD uses half-bankful). This state is expected to "quickly" adjust (it is much shorter memory compared to soil and groundwater). The amount of time required to remove the impact of the initial, fictitious value depends on mainly on river geometry and climate. After additional verifications, we decided to remove the full year of 1950. We added the following sentence to the manuscript (line 169): "*As water volumes at the first time step in the channels are not known, the model sets a conventional initial volume (OS LISFLOOD uses half-bankful), leading to unrealistic initial discharge in some catchments.*"

Line 519 'Factors that can explain the poor performances (..) include the combination of arid climates and the strong influence of lakes and reservoirs': this does not explain performance. Does this mean that the model is worse at simulating arid climates and lakes/reservoirs?

R: Yes, that is exactly what it means.

Line 723-725 'With its refined spatial and temporal resolution, HERA represents hydrological processes in Europe with more detail than previous publicly available hydrological reanalysis products': since there are other hydrological reanalysis available that represent hydrological processes in Europe with an even more refined spatial resolution, remove sentence.

R: We understand the reviewer's suggestion. However, our statement is based on our understanding of the publicly available datasets at the time of writing. If there are indeed other publicly available hydrological reanalysis products for Europe with even more refined spatial resolutions, we would appreciate it if the reviewer could provide us with specific examples or references to these datasets. This would enable us to accurately assess the resolution of HERA in relation to other available products and revise the sentence accordingly.

Line 728-730 'Parameters in 93.5% of the HERA (…)': 'Parameters in 93.5% of the HERA domain (…)'

R: This has been corrected.

Line 730-732 'This is a very high calibration coverage for a GHM, which are not systematically calibrated (…)': many GHMs are systematically calibrated.

R: Models are not systematically calibrated in the following studies:

Beck, H. E., van Dijk, A. I. J. M., de Roo, A., Dutra, E., Fink, G., Orth, R., and Schellekens, J.: Global evaluation of runoff from 10 state-of-the-art hydrological models, Hydrology and Earth System Sciences, 21, 2881–2903, https://doi.org/10.5194/hess-21-2881-2017, 2017.

Hoch, J. M., Sutanudjaja, E. H., Wanders, N., van Beek, R. L. P. H., and Bierkens, M. F. P.: Hyper-resolution PCR-GLOBWB: opportunities and challenges from refining model spatial resolution to 1 km over the European continent, Hydrology and Earth System Sciences, 27, 1383–1401, https://doi.org/10.5194/hess-27-1383-2023, 2023.

Schellekens, J., Dutra, E., Weiland, F. S., Minvielle, M., Calvet, J.-C., Decharme, B., Eisner, S., Fink, G., Flörke, M., Peßenteiner, S., van Beek, R., Polcher, J., Beck, H., Orth, R., Calton, B., Burke, S., Dorigo, W., Weedon, G. P., and Delft, H.: A global water resources ensemble of hydrological models: the eartH2Observe Tier-1 dataset, 2017.

Nevertheless, our original statement indeed may have caused confusion and the point is that our reanalysis is based on many stations in the calibration, so we have rephrased the sentence: "This is a very high calibration coverage for a GHM (Beck et al., 2017), that can be explained by the relatively high coverage in river gauging stations in Europe."

Line 734-738 'It is difficult to compare HERA with other recent hydrological reanalyses (…) for several reasons: (i) spatial coverage (global vs continental), (ii) spatial resolution (0.25º, 0.05º, 1'), (iii) temporal coverage (iv) dynamic vs static socioeconomic conditions': results can still be compared. Since this study presents a dataset the reported performance of the datasets can be compared. See also line 758-760.

R: We appreciate the reviewer's suggestion to compare the reported performances of HERA with other recent hydrological reanalyses, such as ERA5-GLOFAS. However, a direct comparison of the

performance metrics may not be entirely meaningful due to the differences in the validation approaches, catchment sizes, and spatial coverage between the two studies. While it is technically possible to compare the reported performance metrics, such as KGE', Pearson r, bias ratio, and variability ratio (see table below).

Here is a simple comparison of the reported performances between HERA, GLOFAS-ERA5 and the mHM run used for the detailed comparison:

| Dataset | HERA | ERA5-GLOFAS | EUmHM | GRFR |
|---|---|---|---|---|
| Reference | Tilloy et al. (2024) | Harringan et al. (2020) | Samaniego et al. (2019) | |
| Spatial coverage | Europe | Global | Europe | |
| Temporal coverage | 1951-2020 | 1979-Present | 1960-2010 | Global |
| validation catchments (N) | 2848 | 1801 | 357 | 14698 |
| Median validation catchment area (km$^2$) | 583 (27% of catchment area below 250 km2) | 30 046 | 1 700 | Not provided (29% of catchment area below 250 kn2) |
| KGE' (median) | 0.55 (58% > 0.5) | 0.33 | 0.6 | Not provided (27% > 0.5) |
| Pearson r (median) | 0.73 | 0.61 | 0.8 | Not provided |
| Bias ratio (% of catchments with bias ratio between 0.8-1.2) | 50 | 28 | 50 | 44 |
| Variability ratio (% of catchments with variability ratio below 1) | 83 | 61 | 65 | Not provided |

In our opinion, the difference in stations used in the validation in different studies (catchment area, spatial coverage) makes this quick comparison rather meaningless. We argue that a meaningful comparison would be to extract data for the same catchments and compare performances similarly to what has been done with the mHM run. Furthermore, this comparison would be on a small amount of large European catchments, which is less meaningful than the comparison done with the mHM run.

We add the table to the Supplementary material with a short explanatory text and mention it in the main manuscript (line 655 of new TC document).

Lines 770-782: this is all just speculation. Please use the results for both the calibrated and uncalibrated station performance (i.e. uncertainty due to model parameter values) and the mHM comparison (i.e. uncertainty due to model structure), as these are quantified.

R: We modified this part of the discussion in line with your recommendation, the following sentence has been removed as it is indeed quite speculative: "*The improvement of overall modelling performance through time could therefore be related to improving climate inputs, as observations in ERA5-land become sparser and more inhomogeneous as we go back further in time (Hersbach et al., 2020; Muñoz-Sabater et al., 2021)*"

*The following sentences were added to bring more insight about the impact of calibration in skills:*

*Line 684: "In summary, the main strength of HERA lies in its relatively low bias in comparison to the other hydrological datasets considered here (Table S6, Figure S6), while its performances are hampered by its underestimation of variability."*

*Line 704: "Calibration generally improves streamflow simulations (Hirpa et al., 2018) and also HERA shows a better performance for stations used in the calibration process (Figure 7.d). The negative biases and variability ratios can be related to the different meteorological forcing (EMO-1) used in the calibration, although an underestimation of the variability was also found in the EFAS v5.0 run (that is forced by EMO-1). The method, parameters and skill metrics used for calibration further affects model performance. Despite its qualities, the skill metric used for the calibration presented in Section 2.1.2 (KGE') is known to result in an underestimation of variability (Brunner et al., 2021b) and to put more weight on high values (Garcia et al., 2017). This could partly explain the reduced performance in reproducing extreme low flows observed in Figure 8 and Figure 9."*

Line 842-844 'To our knowledge, no other publicly available hydrological reanalysis currently provides discharge data at similar scales and spatiotemporal coverage for Europe.': This study even compares with mHM, which provides discharge data at higher spatial scales and better spatiotemporal coverage (i.e. all cells, not just >100km2) for Europe. Other studies also exist and were provided during the previous review round. Remove sentence.

R: The mHM run used in the comparison has a lower spatial resolution (5kmx5km) than HERA (1.8kmx1.8km), and provides daily data, whereas HERA provides 6-hourly data. So the underlying spatiotemporal resolution of the hydrological modelling is higher compared to any existing product, and therefore is able to capture more of the variability in space and time relevant for hydrological response in catchments. However, for several reasons we decided to make available only data for river pixels with upstream area larger than 100 km2 (corresponding to aggregation of around 30 grid cells in HERA, or 4 in mHM and 1 in GLOFAS ERA5). ). Additionally to reducing storage space, we consider uncertainties being too high at grid level and for very small catchments. We observe a performance decline for smaller catchments, as shown in Figure 7b of the manuscript. The objective of our hydrological run is not to provide an accurate simulation at pixel level, which we argue is not yet possible due to data limitations and conceptualization of catchment processes, but rather to reproduce catchment response at uniform spatiotemporal scale across Europe with reasonable accuracy. We argue that an upstream area of 100 km2 is a fair compromise in this respect.

Line 852-854 'The increased spatial resolution improves the performance due to a better representation of hydrological processes and inputs required to simulate them, including the river network': this conclusion cannot be drawn from this study. The performance increase (compared to what?) could also be due to the changes in inputs this study made.

R: R: This statement is backed with references that also deal with high-resolution GHMs. We are not saying that performance has increased compared to a benchmark, but that increasing the resolution tends to increase performance.

Line 859-862 'The modelling framework developed here further forms a basis for creating alternative (counterfactual) time series of river discharges where climatic or socioeconomic conditions can be kept static, enabling the attribution of changes in hydrological regimes across Europe': indeed, the modeling framework, not HERA itself.

R: We are happy the reviewer agrees with this statement.